# Global climate-related predictors at kilometre resolution for the past and future

Philipp Brun[1], Niklaus E. Zimmermann[1], Chantal Hari[1,2,3,4], Loïc Pellissier[1,5], Dirk Nikolaus Karger[1]

[1] Swiss Federal Research Institute WSL, 8903 Birmensdorf, Switzerland

[2] Wyss Academy for Nature at the University of Bern, 3011 Bern, Switzerland

[3] Climate and Environmental Physics, Physics Institute, University of Bern, 3012 Bern, Switzerland

[4] Oeschger Centre for Climate Change Research, University of Bern, 3012 Bern, Switzerland

[5] Ecosystems and Landscape Evolution, Institute of Terrestrial Ecosystems, Department of Environmental System Science, ETH Zürich, 8092 Zürich, Switzerland.

*Correspondence to*: Philipp Brun (philipp.brun@wsl.ch)

**Abstract.** A multitude of physical and biological processes on which ecosystems and human societies depend are governed by climate, and understanding how these processes are altered by climate change is central to mitigation efforts. We developed a set of climate-related variables at yet unprecedented spatiotemporal detail as a basis for environmental and ecological analyses. We downscaled time series of near-surface relative humidity (*hurs*) and cloud area fraction (*clt*), under the 
consideration of orography and wind, as well as near-surface wind speed (*sfcWind*) using the delta-change method. Combining these grids with mechanistically downscaled information on temperature, precipitation, and solar radiation, we then calculated vapour pressure deficit (*vpd*), surface downwelling shortwave radiation (*rsds*), potential evapotranspiration (*pet*), climate moisture index (*cmi*), and site water balance (*swb*), at a monthly temporal and 30 arcsec. spatial resolution globally, from 1980 until 2018 (time-series variables). At the same spatial resolution, we further estimated climatological normals of frost change 
frequency (*fcf*), snow cover days (*scd*), potential net primary productivity (*npp*), growing degree days (*gdd*), and growing season characteristics for the periods 1981-2010, 2011-2040, 2041-2070, and 2071-2100, considering three shared socioeconomic pathways (SSP126, SSP370, SSP585) and five Earth system models (projected variables). Time-series variables showed high accuracy when validated against observations from meteorological stations, and when compared to alternative products. Projected variables were also highly correlated to observations, although some variables showed notable 
biases, e.g., snow cover days. Together, the CHELSA-BIOCLIM+ data set presented here (https://doi.org/10.16904/envidat.332, Brun et al., 2022) allows improving our understanding of patterns and processes that are governed by climate, including the impact of recent and future climate changes on the world's ecosystems and associated services to societies.

## 1 Introduction

Climate change is impacting multiple facets of the Earth system with consequences on the functioning of natural ecosystems, on the persistence of biological diversity, and on human societies (IPCC, 2022; IPBES, 2018). Climate regulates a broad variety of processes on Earth. It feeds, for example, rivers with precipitation and it generates wind, which is critical for renewable energy production (IPCC, 2011), and it fuels ecosystem and agricultural productivity (Howden et al., 2007), which sustain nearly all life on Earth, including humans (Bellard et al., 2012; Araújo and Rahbek, 2006; Willis and Bhagwat, 2009). Many of these processes react sensibly to climate change (IPCC, 2022) and in order to mitigate negative impacts, a sound understanding of the underlying relationships is key. Among the impacts of climate change are, for example, recent droughts and associated disturbances, such as forest diebacks (Allen et al., 2010), which have fostered studies that attempted to identify and characterise the responsible climate signatures (Seneviratne et al., 2012; Zscheischler et al., 2018). Most of these disruptive events can only be detected and analysed at high spatial and/or temporal resolutions and within a restricted area and period (Easterling et al., 2000). In many regions of the world, existing climate time series lack such high resolution, and thus only to a limited degree allow establishing an understanding of how climate interacts with the natural and human system (Easterling et al., 2016). By the end of the 21$^{st}$ century climate change is expected to lead to profound changes in the distribution ranges of species and ecosystems (Thuiller et al., 2005, 2019). A reasonable anticipation of such changes must rely on sound information on climate-related variables, considering different climate-change scenarios at an informative spatial and temporal resolution. The availability of relevant climate-related data at high spatiotemporal resolution for current conditions and for the decades ahead of us is therefore crucial to fill the gaps in our understanding of climate-change impact on the Earth system.

A popular repository for climate data is hosted by the climatologies at high resolution for the Earth's land surface areas (CHELSA) initiative (Karger et al., 2017, 2020, 2021b), which provides information on temperature and precipitation globally at kilometre resolution. Originally, the CHELSA initiative offered climate data primarily as climatologies, i.e., as monthly and seasonal statistics typically averaged over a representative period of 30 years or longer (Arguez and Vose, 2011), initially from 1979-2013. A key set of such climatologies are the 19 bioclimatic variables (Hijmans et al., 2005) that represent seasonal and annual statistics of precipitation and temperature and are widely used as predictors in macroecology (Fourcade et al., 2018). However, while these original data may be relevant for many applications, they have three primary limitations: they only (1) include variables that independently summarise either temperature or precipitation, (2) represent long-term climatic conditions, and (3) represent the recent past.

For a sound understanding of how physical and biological processes are driven by climate, information on temperature and precipitation alone is not sufficient. Assessing the potential for solar energy production, for example, is impossible without knowing how much shortwave solar radiation (*rsds*) reaches a location of interest. Similarly, precipitation may measure the amount of water that reaches the surface, but across the globe this is an inaccurate proxy for the amount of water that is

available to plants: 300 kg m$^{-2}$ annual precipitation, for instance, can be found in the Alaskan taiga, in the Mongolian steppe, or in the Pakistani desert (Karger et al., 2017), where the dominant vegetation exhibits large differences in the ability to cope with water stress. Across these systems a much more accurate indicator of water stress is the climate moisture index (*cmi*, Hogg, 1997), i.e., the difference between precipitation and potential evapotranspiration (*pet*), as *pet* differs by a factor of three between the Alaskan taiga and the Pakistani desert (Singer et al., 2021). The popularity of the 19 bioclimatic variables to summarise climate therefore appears to result rather from the lack of relevant alternatives with kilometre-resolution than from their imminent relevance.

Time-series data on climate-related variables are indispensable to understand the drivers of the many important Earth system processes that vary with time. Resolving how the primary weather patterns unfold, for example, allows for a much deeper understanding of the control of spatiotemporal patterns of ecosystem productivity (Hartman et al., 2020). Similarly, time-series of *pet* and *cmi* can be used to understand the country-wide temporal dynamics in crop yield (Zhang et al., 2015; Santini et al., 2022). Modelling crop yields based on sound *pet* and *cmi* data may, in turn, allow for a better anticipation of shortages in food production and agricultural planning. Moreover, extreme weather anomalies such as droughts can be identified at large scales and better linked to consequential disturbances like wildfires and forest diebacks. While for temperature and precipitation such time series of high temporal (daily) resolution data have recently been published (Karger et al., 2020, 2021b), global time-series at kilometre resolution are hardly available for additional climate-related variables relevant to understand ecosystems processes.

In order to anticipate and mitigate the manifold impacts of climate change until the end of this century, future projections of meaningful climate-related variables are required. Climate change is expected to continue or even accelerate in the coming decades and its impacts on ecosystems and human societies are likely becoming stronger (IPCC, 2022). Crop yields, for example, are expected to change, tracking their optimal climate (Leng and Hall, 2019; IPCC, 2022): in high latitudes, harvests may become bigger due to warming, whereas elsewhere irrigation may become necessary to keep growing traditional crops (Liu et al., 2021; Masia et al., 2021). In certain areas some crops will likely have to be abandoned entirely and replaced with better adapted alternatives (Sloat et al., 2020). Such agricultural system changes are costly, take time, and are only efficient if the expected changes can be reasonably-well anticipated. Similarly, coping with the ongoing biodiversity crisis requires a rapid establishment of an optimally-designed global network of protected areas (Elsen et al., 2020; Hannah, 2008; Pollock et al., 2017). Yet, finding the most sustainable way to create such a network requires knowledge on the expected changes in climate and their impacts on the distribution ranges of species. For temperature and precipitation, high-resolution future climatologies have been made available (Karger et al., 2017), but this is generally not the case for other climate-related variables that are more directly linked to ecosystem processes.

Here, we present the CHELSA-BIOCLIM+ (climatologies at high resolution for the Earth's land surface areas – bioclimatic variables plus) dataset of global kilometre-resolution time series and climatologies for 15 climate-related variables. We compiled input data from CHELSA V.2.1 (Karger et al., 2021a) and other high-quality sources and used state-of-the art approaches to generate two groups of biologically relevant climate-related variables: for one group of variables we created time-series covering 39 years of the recent past (hereafter: time-series variables), and for the other group we created climatologies for current and expected future conditions (hereafter: projected variables). Time-series variables are available for the period of 1980-2018 and include near-surface relative humidity (*hurs*), cloud area fraction (*clt*), near-surface wind speed (*sfcWind*), vapour pressure deficit (*vpd*), surface downwelling shortwave (solar) radiation (*rsds*), potential evapotranspiration (*pet*), and climate moisture index (*cmi*), each of which containing 468 monthly layers at a 30 arcsec resolution (i.e., less than 1 km), and an annual statistic, i.e., site water balance (*swb*), containing 38 annual layers. For all of these variables but site water balance, we further calculated climatologies monthly, annually, and for annual ranges and extrema for the period 1981-2010, which is the climate-normal period recommended by the World Meteorological Organization (Arguez and Vose, 2011). Projected variables include frost change frequency (*fcf*), snow cover days (*scd*), potential net primary productivity (*npp*), growing degree days (*gdd*), growing season length (*gsl*), growing season temperature (*gst*), and growing season precipitation (*gsp*), for which we calculated climatological means for the same kilometre-resolution grid for the periods 1981-2010, 2011-2040, 2041-2070, and 2071-2100. For the latter three periods, climatological values were generated for each combination of three shared socio-economic pathways (SSPs, O'Neill et al., 2014) and five Earth system models. To demonstrate the robustness of these variables, we validated them, where feasible, against global sets of observations from meteorological stations and we compared them with existing products. Together, our layers of climate-related variables allow the characterisation of each pixel of the life-supporting landmass on Earth far more comprehensively than would be possible from temperature and precipitation alone: for the recent decades with monthly resolution and until the end of this century as projected climatologies.

## 2 Material and Methods

We developed 15 climate-related variables that complement and build on existing products of the CHELSA initiative (Karger et al., 2017). We classified these variables into five orders, representing increasing degrees of abstraction from *in situ* measurements (Fig. 1). First-order variables are directly measurable properties, including near-surface temperature (daily means and extrema), precipitation rates, near-surface relative humidity, cloud area fraction, solar radiation, and near-surface wind speed. While downscaled climatologies and time-series of temperature and precipitation rates have been made available

previously (Karger et al., 2017, 2020, 2021b), here we downscaled corresponding layers for the remaining first-order variables total cloud cover (*clt*), near-surface (10m) wind speed (*sfcWind*), and near-surface relative humidity (*hurs*). Directly based on these first-order variables, we have generated time series and climatologies for five biologically meaningful second-order variables, including frost change frequency (*fcf*), snow cover days (*scd*), potential net primary productivity (*npp*), and vapour pressure deficit (*vpd*). In addition, we aggregated daily high-resolution time series of surface downwelling shortwave radiation

(*rsds*), that have been developed in a related study (Karger et al. in prep.). Similarly, we have generated time series and climatologies of four third-order climate variables (based on first and second order variables), including growing season length (*gsl*), growing season precipitation (*gsp*), and growing season temperature (*gst*), and potential evapotranspiration (*pet*), as well as one fourth and one fifth order variable, i.e., climate moisture index (*cmi*) and site water balance (*swb*), respectively.

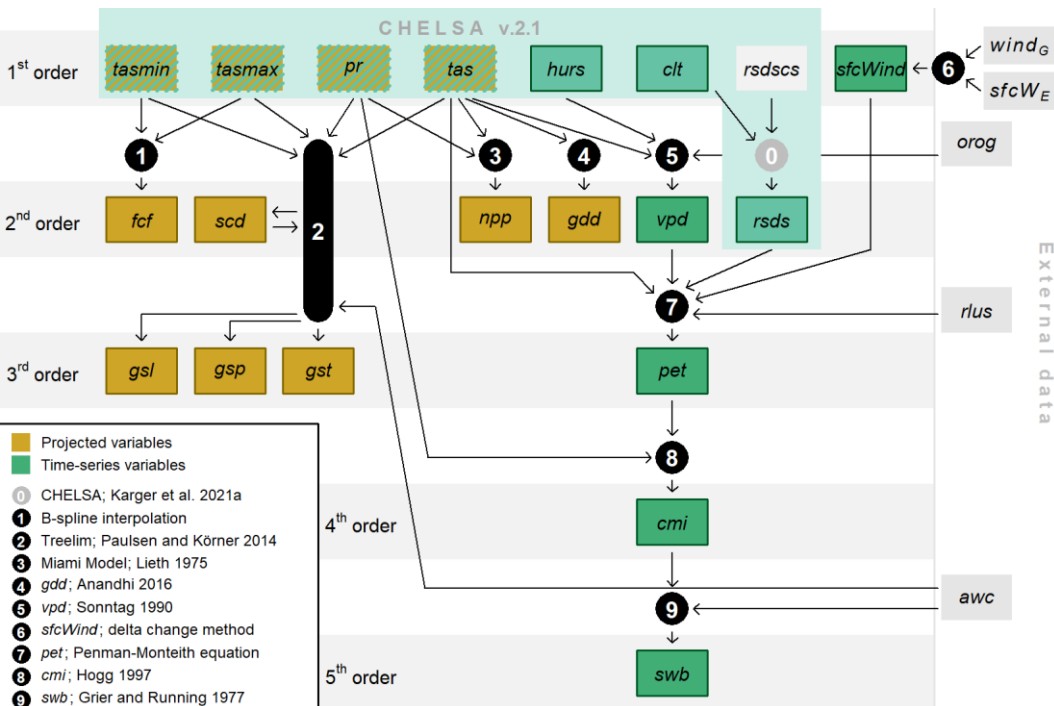

**Figure 1: Input data, analyses, and output variables generated. *tasmin* represents daily minimum near-surface air temperature; *tasmax* represents daily maximum near-surface air temperature; *pr* represents precipitation rates; *tas* represents near-surface daily average air temperature; *hurs* represents near-surface relative humidity; *clt* represents cloud area fraction; *rsdscs* surface downwelling shortwave radiation assuming clear sky; *orog* represents orography; *fcf* represents frost change frequency; *scd***

represents snow cover days; *npp* represents potential net primary productivity; *gdd* represents growing degree days; *vpd* represents vapour pressure deficit; *rsds* represents surface downwelling shortwave radiation corrected for atmospheric transmissivity and topography; *sfcWind* represents near-surface wind speed; *sfcW$_E$* represents near-surface wind speed from ERA5; *wind$_G$* represents wind speed from Global Wind Atlas; *rlus* represents surface upwelling longwave radiation; *gsl* represents growing season length; *gsp* represents growing season precipitation; *gst* represents growing season temperature; *pet* represents potential evapotranspiration; *cmi* represents climatic moisture index; *awc* represents available soil water capacity; and *swb* represents site water balance. Green squares represent climate variables for which monthly time series are available for the period 1980-2018; orange squares represent variables for which future projections of climatologies exist; hashed squares represent variables with both time series for the recent past and future projections. Squares with border lines are part of the data set presented.

## 2.1. Input data

Data on near-surface air temperature (*tasmin, tasmax, tas*), as well as precipitation rates (*pr*) and surface downwelling shortwave radiation (*rsds*) have been taken from CHELSA V2.1 (Karger et al., 2021b). For past conditions, forcing from ERA5 (Hersbach et al., 2020) with a GPCC bias correction (Ziese et al., 2018) was used, as well as an air temperature algorithm that builds on an atmospheric lapse rate-based downscaling (Karger et al., 2017). Precipitation rates (*pr*) are based on a mechanistic downscaling that takes orographic effects into account (Karger et al., 2021b). Surface downwelling shortwave radiation (*rsds*) in CHELSA V2.1 is based on a terrain-specific, mechanistic model (Böhner and Antonic, 2009). For *tasmin*, *tasmax*, *tas*, and *pr* we also used data on projected monthly climatologies for the periods 2011-2040, 2041-2070, and 2071-2100 from CHELSA V2.1. Such climatologies were generated for three official SSPs (O'Neill et al., 2016, 2017): SSP126 is an optimistic emission scenario, assuming that the world shifts gradually to a more sustainable path, resulting in additional radiative forcing of 2.6 W m$^{-2}$ by 2100, relative to preindustrial levels; SSP370 is an intermediate-to-pessimistic scenario, assuming that international fragmentation and regional rivalry hamper efficient implementations of globally sustainable solutions, leading to additional radiative forcing of 7.0 W m$^{-2}$ by 2100; SSP585 is a pessimistic emission scenario, assuming that developing countries follow the trajectories of first world countries in rapid economic development that hardly relies on greenhouse gas-efficient technologies. It assumes additional radiative forcing of 8.5 W m$^{-2}$ by 2100. For each of these SSPs, we used global simulations of five Earth system models that were prepared for the Intersectoral Impact Model Intercomparison Project round 3b (ISIMIP3b, https://www.isimip.org/) to generate future climatic anomalies of precipitation and temperature. Earth system models were chosen based on the availability of all needed climate variables and model performance following ISIMIP3b (Lange, 2021) and included GFDL-ESM4 (Held et al., 2019), IPSL-CM6A-LR (Boucher et al., 2020), MPI-ESM 1-2-HR (Gutjahr et al., 2019), MRI-ESM2-0 (Yukimoto et al., 2019), and UKESM1-0-LL (Sellar et al., 2019). In a first step, for each variable (*tasmin*, *tasmax*, *tas*, and *pr*) the dynamic model outputs were used to generate monthly climatologies for the 3 periods × 3 SSPs × 5 Earth system models. In addition, for each Earth system model and climate variable, one climatology was generated for the period 1981-2010. Then, each of these climatologies was downscaled to 30 arcsec using the delta-change method (Hay et al., 2000).

In addition, we compiled data for orography (*orog*), wind speed (*wind$_G$*, *sfcW$_E$*), relative humidity (*hur*), total cloud cover (*tcc*), surface upwelling longwave radiation (*rlus*), and available soil water content (*awc*). Orography data originated from the

Global Multi-resolution Terrain Elevation Data 2010 (GMTED2010; Danielson and Gesch, 2011) at a resolution of 30 arcsec. We obtained two types of wind speed data: long-term averages at high spatial resolution (9 arcsec), and monthly time-series at coarser spatial resolution. Wind speed averages for the period 2008-2017 at ten ($windG_{10}$) and 50 ($windG_{50}$) metres above surface were obtained from the Global Wind Atlas 3.0, a free, web-based application developed, owned and operated by the Technical University of Denmark (https://globalwindatlas.info). From these layers, we derived roughness length as

$$z_0 = e^{\frac{windG_{10}ln(50)-windG_{50}ln(10)}{windG_{10}-windG_{50}}}, \tag{1}$$

Then, we aggregated $windG_{10}$ and roughness length from the original 9 arcsec resolution to 30 arcsec, using a two-step approach. First, we aggregated to 27 arcsec (factor of three) by median, and then we resampled to 30 arcsec, using cell area-weighted means. Finally, in order to keep aggregated roughness length estimates in a realistic range and to remove a few outliers, we bounded them by the typical values for the open sea (0.0002) as minimum and city centres with skyscrapers/mountain tops (4) as maximum (WMO, 2018). Monthly time-series of wind speed 10 m above the surface were obtained from the ERA5 global reanalysis product ($sfcW_E$, Hersbach et al., 2020) released by the European Centre for Medium-Range Weather Forecasts (ECMWF), and covered the period 1979-2020 with a horizontal resolution of 0.25°. From ERA5, we also used relative humidity (*hur*) and total cloud cover (*tcc*) at 0.25° resolution monthly for the period 1980-2018. Monthly information on the surface upwelling longwave radiation needed for the calculation of *pet* was obtained from the ERA5-Land reanalysis product (Muñoz Sabater, 2019; Muñoz-Sabater, 2021) that is also maintained by the ECMWF. It covered the period 1979-2020 with a horizontal resolution of 0.1°. Information on available soil water capacity (*awc*) was obtained from SoilGrids (Hengl et al., 2014, 2017) with a horizontal resolution of 30 arcsec and a vertical resolution of six soil layers. From these data, we calculated one layer of available water volume by integrating over the soil profiles. A summary of all input data used is provided in Table 1.

**Table 1: Input data used to generate the CHELSA-BIOCLIM+ data set**

| Variable name | Description | Spatial resolution | Temporal resolution | Time period | Source |
|---|---|---|---|---|---|
| *tasmin* | Daily minimum near-surface air temperature | 30 arcsec | monthly | 1979-2019 | Karger et al. 2017 |
| | | 0.5° | | 2011-2100 | ISIMIP3b |
| *tasmax* | Daily maximum near-surface air temperature | 30 arcsec | monthly | 1979-2019 | Karger et al. 2017 |
| | | 0.5° | | 2011-2100 | ISIMIP3b |
| *tas* | Mean near-surface air temperature | 30 arcsec | monthly | 1979-2019 | Karger et al. 2017 |
| | | 0.5° | | 2011-2100 | ISIMIP3b |
| *pr* | Precipitation rate | 30 arcsec | monthly | 1979-2018 | Karger et al. 2017 |
| | | 0.5° | | 2011-2100 | ISIMIP3b |

| | | | | | |
|---|---|---|---|---|---|
| *rsds* | Surface downwelling shortwave radiation | 30 arcsec | daily | 1980-2018 | Karger et al. in prep. |
| *orog* | Orography | 30 arcsec | - | - | Danielson and Gesch (2011) |
| *hur* | Relative humidity | 0.25° | monthly | 1980-2018 | ERA5 |
| *tcc* | Total cloud cover | 0.25° | monthly | 1980-2018 | ERA5 |
| *sfcWE* | Wind speed at 10 m above the surface | 0.25° | monthly | 1979-2020 | ERA5 |
| *rlus* | Surface upwelling longwave radiation | 0.1° | monthly | 1979-2020 | ERA5-Land |
| $windG_{10}$ | Wind speed at 10 m above the surface | 9 arcsec | - | 2008-2017 | Global Wind Atlas 3.0 |
| $windG_{50}$ | Wind speed at 50 m above the surface | 9 arcsec | - | 2008-2017 | Global Wind Atlas 3.0 |
| *awc* | Available soil water capacity | 30 arcsec | - | - | SoilGrids |

## 2.2 Generating raster layers

### 2.2.1 First-order climate layers

**Near-surface relative humidity (*hurs*)**

Near-surface relative humidity (*hurs*) controls the biologically important variable vapour pressure deficit (see below) as well as fog formation (at *hurs* = 100 %), which can be a critical water source for vegetation in certain coastal ecosystems (e.g., in the California redwood forest; Dawson, 1998). We calculated *hurs* from atmospheric *hur* at pressure levels *z*. We used all pressure levels from ERA5 and horizontally B-spline ($S_{xy}$) interpolated *hur* at pressure levels $z_{i=1}...z_n$ to a 30 arcsec resolution, using longitude (*x*) and latitude (*y*) as predictors and *hur* as response, so that:

$$S_{xy}(hur) = f(x, y) \tag{2}$$

From the resulting spline-interpolated values $S_{xy}(hur)$ for each pressure level *z*, we then calculated a vertical spline interpolation separately for each 30 arcsec grid cell, using the geopotential height of each layer divided by the gravitational constant $g=9.80665$ m s$^{-2}$ as predictor, and the values given by the function $S_{xy}(hur)$ as response so that:

$$S_z(hur) = S\left(S_{xy}(hur)\right) = f(z) \tag{3}$$

We then used the vertical spline $S_z(hur)$ to calculate a first approximation of $hurs_{orog}$ at the 30 arcsec, with $orog$ referring to

surface elevation. This first approximation of the relative humidity at the surface, however, does not include orographic effects such as increased $hurs$ at windward and lower $hurs$ on leeward sides of an orographic barrier. Moist air is raising on the windward side of an orographic barrier, potentially losing moisture and cooling with a wet-adiabatic lapse rate, and sinking on its leeward side, usually warming with a higher, dry-adiabatic lapse rate. This effect of differing adiabatic lapse rates and consequently temperature changes affects relative humidity. To include these orographic effects into the estimation of $hurs$,

we use:

$$hurs = \frac{1}{(1+exp(-1\cdot h))}, \tag{4}$$

with

$$h = \frac{h_t \cdot \left(H + (H_c - H)(1 - H_c)\right)}{H_c}, \tag{5}$$

and $h_t$ being the *logit*-transformed version of $hurs_{orog}$:

$$h_t = log\left(\frac{hurs_{orog}}{1 - hurs_{orog}}\right), \tag{6}$$

$H$ being the windward leeward index at 30 arcsec resolution calculated following the same parametrization as used in Karger et al., (2021b), and $H_c$ being the spline-interpolated mean of all $H$ values that overlap with the respective 0.25° grid cell from ERA5. We calculated $hurs$ monthly for the period 1980-2018. For the period 1981-2010, we derived monthly climatologies

and climatological means, annual ranges and extrema. All $hurs$ data are reported as percentages.

**Cloud area fraction (*clt*)**

The cloud area fraction (*clt*) represents the fraction of a grid cell that is covered by clouds across the entire atmospheric column, as seen from the Earth's surface or the top of the atmosphere. It includes both large-scale and convective clouds. Cloud area

fraction determines the amount of downwelling solar radiation that reaches the Earth's surface and is an important constraint to productivity in tropical ecosystems (Nemani et al., 2003). Moreover, low-hanging clouds can be a key water source, and thus in mountain regions *clt* can be an important determinant of the distribution of tropical cloud forests (Karger et al., 2021c). We calculated *clt* monthly for the period 1980-2018 based on *tcc,* and following the procedure described in (Karger et al.,

submitted to ESSD). Unlike all other variables presented here, we downscaled *clt* to a cruder spatial resolution of 1.5 arcmin.
This resolution was chosen, because it is similar to the resolution at which orographic wind effects for precipitation are calculated (Karger et al., 2021b), and because it avoids over-representing terrain effects (Daly et al., 1997). For the period 1981-2010, we derived monthly climatologies and climatological means, annual ranges and extrema. All *clt* data are reported as percentages.

**Near-surface wind speed (*sfcWind*)**

Numerous direct and indirect effects of wind speed on terrestrial ecosystems exist, including gas and heat exchange, dispersal of pollen, seeds, pests or pollutants, and wind throw (Nobel, 1981). The impact of wind exposure on microclimate and vegetation patterns are particularly evident, for example, in the polar and subpolar zones (Schultz, 2005). We estimated monthly averages of near-surface (10 m) wind speed (*sfcWind*) at 30 arcsec resolution by downscaling and bias-correcting the ERA5 time-series (*sfcW_E*), using an aggregation of the Global Wind Atlas product ($windG_{10}$; see subsection Input data). In a first step, we averaged *sfcW_E* for the period 2008-2017, for which the Global Wind Atlas is representative. Then, we estimated the average deviation between $sfcW_E$ and $windG_{10}$. This deviation raster contained information about both small-scale deviations from the ERA5 cell mean due to topography and bias in long-term estimates of wind speed. Next, we added this difference layer to each monthly ERA5 layer (from 1979-2019), after log-transforming all layers. Our approach therefore corresponded to the delta-change method (Hay et al., 2000), except that we applied it on log-transformed wind speed estimates. This was done because wind speed follows a Weibull distribution (Weibull, 1951), which can be related to the normal distribution through a log link function. Finally, we back-transformed the two-layer sums by exponentiating them. For the period 1981-2010, we derived monthly climatologies of *sfcWind* and climatological means, annual ranges and extrema. All *sfcWind* data are reported in metres per second.

**2.2.2 Second-order climate layers**

**Frost change frequency (*fcf*)**

Frost change frequency (*fcf*) describes the number of days per year with temperature minima below 0 °C and maxima above 0 °C. Coping with frost requires adapted behaviours or elaborate physiological adaptations for both ecto- and endothermal organisms, and especially for non-migrating life forms that cannot escape, such as plants. Frost change frequency carries information about the occurrence frequency of freezing and thawing events and - indirectly - about their duration, both of which are crucial constraints determining the best-suited adaptation strategies, see e.g., Hufkens et al. (2012). We used a B-spline interpolation $S(tasmax, t)$ and $S(tasmin, t)$ to get both daily minimum ($tasmin_i$) and maximum ($tasmax_i$) near-surface 2 m air temperatures from monthly values, with *t* the sequence of Julian days marking the middle of each month, i.e., [349,15,45,74,105,135,166,196,227,258,288,319,349,15]. As B-spline interpolations cannot predict values outside their bounding knots, we first extended the sequence of knots to start at Dec. 15 (Julian day 349), and end at Jan. 15 (Julian day 15), and cut the interpolated sequence to range from Jan. 1 and Dec. 31 in a second step. A frost change event was then defined by

$tasmin_i < 0$ °C and $tasmax_i > 0$ °C. We calculated *fcf* from the monthly climatologies of *tasmin* and *tasmax* for the periods 1981-2010, 2011-2040, 2041-2070, and 2071-2100 for all combinations of SSPs and Earth System models (see subsection Input data). *fcf* is reported as the number of days per year with frost change events.

**Snow cover days (*scd*)**

Snow cover days (*scd*) are the number of days per year on which the ground is covered with snow. Snow cover affects local climate, hydrology, and ecosystems in complex ways (Callaghan et al., 2011; Schultz, 2005), by insulating the soil from temperature minima during winter months (Zhang, 2005), by determining Arctic vegetation patterns (Evans et al., 1989), or by providing hiding opportunities form predators for small mammals (Callaghan et al., 2011). We used a B-spline interpolation $S(tas, t)$ to get from monthly to daily estimates of *tas*, with *t* being a vector of Julian days marking the middle of each month, i.e., [349,15,45,74,105,135,166,196,227,258,288,319,349,15], and *tas* being the mean of near-surface 2 m air temperature for the respective month. We used a stepwise interpolation of monthly precipitation rates to daily precipitation rates following (Paulsen and Körner, 2014). The daily precipitation rate (*pr*) in this approach is directly coupled to the near-surface air temperature as follows:

$$pr = \begin{cases} 5kg \cdot m^{-2} \cdot day^{-1} & if\, tas < 5°C \\ 10kg \cdot m^{-2} \cdot day^{-1} & if\, 5°C \leq tas < 10°C \\ 15kg \cdot m^{-2} \cdot day^{-1} & if\, 10°C \leq tas < 15°C \\ 20kg \cdot m^{-2} \cdot day^{-1} & if\, 15°C \leq tas \end{cases} \tag{7}$$

The total amount of *pr* is distributed to as many rainfall events as are necessary to obtain the monthly amount of precipitation, with events being evenly distributed across the month. Precipitation is solid (snow) when $tas < 0$ °C and accumulates as long as *tas* remains below 0 °C. If $tas > 0$ °C it melts by a rate of 0.84 kg m$^{-2}$ day$^{-1}$ K$^{-1}$ (Paulsen and Körner, 2014). When liquid precipitation falls on an existing snow layer, it cools to 0 °C and the thermal energy released (4.186 kJ kg$^{-1}$ K$^{-1}$) is assumed to melt snow (Körner et al., 2011). The number of snow cover days (*scd*) is then given by the days of the year on which a snow layer with a snow water content of $\geq$ 1 kg m$^{-2}$ existed. We calculated *scd* from the monthly climatologies of *tas* and *pr* for the periods 1981-2010, 2011-2040, 2041-2070, and 2071-2100 for all combinations of SSPs and Earth System models (see subsection Input data). *scd* is reported as the number of days per year with snow cover.

**Potential net primary productivity (*npp*)**

Potential net primary productivity (*npp*) is the potential difference between the rate at which carbon is fixed by photoautotrophs and the rate at which carbon is emitted through cell respiration, if only climate was limiting. Primary productivity is the main way through which carbon dioxide is removed from the atmosphere and biomass is produced and thus a key ecosystem function (Schimel, 1995). Here, we used the Miami model (Lieth, 1975) to estimate *npp* solely based on climatic constraints, resulting in a potential estimate that is independent of the existing vegetation on the ground. The unit of *npp* is given as *g* m$^{-2}$ yr$^{-1}$, where *g* stands grams of dry matter. The estimates are based on mean annual near-surface 2 m air temperature in °C and annual

precipitation rates in kg m$^{-2}$ yr$^{-1}$. The Miami model assumes that *npp* increases asymptotically with both increasing temperature and increasing precipitation, approaching an upper limit of 3000 *g* m$^{-2}$ yr$^{-1}$. The precipitation component to *npp* is given as

$$npp_{pr} = 3000 \times \left(1 - exp(-0.000664 \times pr)\right),$$ (8)

and the air temperature component is given as

$$npp_{tas} = 3000 \times (1 + exp(1.315 - 0.119 \times tas))^{-1}$$ (9)

Based on these two components, *npp* is either limited by temperature or precipitation, and determined by the minimum estimate of *npp* from either the temperature or the precipitation component:

$$npp = min(npp_{tas}, npp_{pr})$$ (10)

We calculated *npp* from the monthly climatologies of *tas* and *pr* for the periods 1981-2010, 2011-2040, 2041-2070, and 2071-2100 for all combinations of SSPs and Earth System models (see subsection Input data).

**Growing degree days (*gdd*)**

Growing degree days (*gdd*) are a measure of heat accumulation over a specific time period. It has been used to understand the

320 phenology of plants and animals for centuries in agronomy (Anandhi, 2016), and for a shorter period in ecology (Cayton et al., 2015). It has been shown that the heat sum above a critical threshold accumulated through time better explains e.g. plant phenology than a threshold temperature alone (Larcher, 1994). The *gdd* threshold temperature ascertains that cool periods, during which phenological progress stagnates, are omitted. The threshold temperature is species-specific, and varies e.g. between 0 °C for cold-adapted plants (Larcher, 1994) to 5 or 5.5 °C for many temperate to boreal tree species (Prentice et al.,

1992; Lenihan, 1993), while tropical plants are limited by temperatures below 10 °C and even much higher (Larcher, 1994). Growing degree days are calculated by first assessing whether daily mean near-surface 2 m air temperatures surpass a baseline threshold temperature $tas_b$ (e.g., 5 °C), and then summing all the surpluses. To obtain daily estimates of near-surface 2 m air temperature from monthly values we have used the same approach of B-spline interpolation as for snow cover days. The growing degree sum is then given as the sum:

$$gdd_b = \sum_{i=1}^{365}(max\,(tas_i - tas_b, 0)),$$ (11)

where $tas_b$ is the baseline temperature and *i* represents Julian day. We calculated *gdd* for three baseline temperatures (0 °C, 5 °C, and 10 °C) from the monthly climatologies of *tas* for the periods 1981-2010, 2011-2040, 2041-2070, and 2071-2100 for all combinations of SSPs and Earth System models (see subsection Input data). However, here we only report the results for *gdd* with the 5 °C baseline (*gdd₅*). All *gdd* data are reported as degree days (°C day).

**Vapour pressure deficit (*vpd*)**

Vapour pressure deficit (*vpd*) is the difference between the actual amount of moisture in the air and the maximum amount of moisture the air can hold at a given temperature. *vpd* is a key meteorological property for terrestrial biomes, determining plant functioning and drought-induced mortality (Grossiord et al., 2020). Moreover, the distributions of animals prone to dessication such as small arthropods are limited by *vpd* (Hauser et al., 2018; Ouisse et al., 2016). Near-surface *vpd* can be calculated from near-surface relative humidity (*hurs*), considered as unitless fraction, and near-surface air temperature (*tas*) in °C as

$$vpd = e_{sat}(tas) \times (1 - hurs),$$ (12)

where $e_{sat}(tas)$ is the saturation vapour pressure. In order to approximate $e_{sat}(tas)$, we used the Magnus equation with the coefficients of Sonntag (Sonntag, 1990),

$$e_{sat}(tas) = 0.6112 \times e^{\frac{17.62*tas}{(243.12+tas)}},$$ (13)

*vpd* was calculated in the R environment (R Development Core Team, 2008), using the package bigleaf (Knauer et al., 2018). We calculated vapour pressure deficit monthly for the period 1980-2018. For the period 1981-2010, we derived monthly climatologies and climatological means, annual ranges and extrema. All *vpd* data are reported in Pascal (Pa).

**Surface downwelling shortwave radiation (*rsds*)**

Surface downwelling shortwave radiation (*rsds*) is the amount of direct and diffuse shortwave radiation that reaches the Earth's surface, considering the filtering effects of air and clouds throughout the atmosphere, as well as the effects of the local topography. *rsds* describes the amount of solar energy available. It can critically affect local climate and vegetation patterns in high-latitude environments (Andrade et al., 2018; Schultz, 2005). In the tropics with year-round rain, where temperature and precipitation are not limiting, it can constrain primary productivity (Nemani et al., 2003). To calculate *rsds*, surface downwelling solar radiation under clear sky conditions (*rsdscs*) is first calculated by computing 30-arcsec clear-sky radiation using the method described in (Böhner and Antonic, 2009) for each day of the year. Then daily estimates of *rsdscs* and *clt* are combined through the following relationship:

$$rsds = rsdscs \cdot 1 - 0.75 \cdot clt^{3.4}$$ (14)

This way, daily estimates of *rsds* from 1980 to 2018 were generated in a related project (Karger et al., submitted to ESSD). Here, we summarised these estimates to monthly means, and for the period 1981-2010 we derived monthly climatologies and climatological means, annual ranges and extrema. All *rsds* data are reported as MJ m$^{-2}$ day$^{-1}$.

**2.2.3 Third-order climate layers**

**Growing season-related predictors**

The growing season is the annual period, during which conditions are favourable for vegetation growth. Growing season length (*gsl*) indicates the amount of time available for plant growth, which is an important determinant of life-history traits and

productivity (Paulsen and Körner, 2014). As *gdd*, *gsl* is species specific and can vary considerably between plants adapted to different biomes. Here, we estimate *gsl* for tree species forming treelines, i.e., growing at the cold/dry boundary of forested biomes worldwide. Under such conditions *gsl* can be defined as the number of days per year with temperatures > 0.9 °C, with no snow cover being present, and with sufficient water available in the soil (Paulsen and Körner, 2014). Daily precipitation rates and near-surface 2 m air temperature averages were calculated in the same way as for snow cover. In addition, potential evapotranspiration was estimated, using the Hargreaves equation and *tasmin* and *tasmax* as input (Hargreaves and Samani, 1985). Note that this estimate of *pet* is specific to the estimate growing season-related predictors and independent of the more sophisticated approach presented below. Water balance in the soil was calculated by a two-layer bucket model. The upper layer is assumed to be able to hold 30 kg of liquid water per square metre at maximum. For the lower layer we used empirical data on water holding capacity *awc* (see Input data). Liquid precipitation or snow melt fills the upper layer first. If the soil water content of the upper layer ($swc_1$) exceeds 30 kg m$^{-2}$, water flows to the lower layer until saturated. If the second layer is saturated, the remaining flux is assumed to be lost as runoff. If water is present in the upper layer, actual evapotranspiration (*aet*) is equal to *pet*. We used a square-root correction for the estimation of the actual daily evapotranspiration from deeper layers as soon as the upper layer was empty: $aet = pet \times (swc_2/awc)^{1/2}$ in kg m$^{-2}$ day$^{-1}$, with soil water given in kg m$^{-2}$. A growing season day is defined as a day on which $swc_1 > 0$ & $tas > 0.9$ °C & snow < 1 kg m$^{-2}$. Growing season length is then the number of days per year on which this condition holds true, growing season precipitation (*gsp*) is the amount of precipitation that falls during the days on which the condition is true, and growing season temperature (*gst*) is the mean near-surface air temperature during days on which the condition is true. We calculated *gsl*, *gsp*, and *gst* from the monthly climatologies of *tasmin*, *tasmax*, *tas*, *pr*, and from the *scd* estimates described above for the periods 1981-2010, 2011-2040, 2041-2070, and 2071-2100 for all combinations of SSPs and Earth System models (see subsection Input data).

**Potential evapotranspiration (*pet*)**

Potential evapotranspiration (*pet*) is defined as the amount of water per area and time that could evaporate at the soil surface or be transpired through plants if soil water availability was not limiting. Evapotranspiration is a crucial part of the water cycle and strongly interacts with vegetation traits such as leaf area (Irmak, 2008). We calculated *pet* with the Penman-Monteith equation (Monteith, 1965) as implemented in the R package bigleaf (function 'potential.ET'). This function builds on the following equation (Knauer et al., 2018):

$$\lambda E_{pot} = \frac{\Delta(R_n - G - S) + \rho \times c_p \times vpd \times Ga}{\Delta + \gamma\left(1 + \frac{Ga}{Gs_{pot}}\right)},$$ (15)

where $\Delta$ is the slope of the saturation vapour pressure curve [kPa K$^{-1}$] that is approximated with equation 3; $R_n$ is net radiation [W m$^{-2}$]; $G$ is the ground heat flux [W m$^{-2}$]; S is the sum of all storage fluxes [W m$^{-2}$]; $\rho$ is the mean air density [kg m$^{-3}$]; $c_p$ is the specific heat of the air [J K$^{-1}$ kg$^{-1}$]; $\gamma$ is the psychrometric constant [kPa K$^{-1}$], $Ga$ is the aerodynamic conductance [m s$^{-1}$]; and $Gs_{pot}$ is the potential surface conductance [mol m$^{-2}$ s$^{-1}$]. To calculate *pet* with the bigleaf framework, information on the

following general environmental conditions is required: *tas*, *vpd*, $R_n$, pressure, *G*, and *S*. For *tas*, we used monthly layers of the CHELSA *tas* product (see Input data). For *vpd*, we used the layers calculated here. $R_n$ was calculated as the difference between surface downwelling shortwave radiation (*rsds*) calculated here, and surface upwelling longwave radiation (*rlus*) from ERA5-Land, following (Singer et al., 2021). Since these radiation layers had different spatial resolutions (30 arcsec and 0.1°, respectively), we used the grid calculus tool of the System for Automated Geoscientific Analyses (SAGA, Conrad et al., 2015) to calculate the differences on the fine grid, using bilinear interpolation to downscale the coarse grid of *rlus*. In a few pixels (in rugged terrain) estimates of $R_n$ could be negative, in which case we manually set them to zero. Pressure was calculated with the function 'pressure.from.elevation' of the R package bigleaf (Knauer et al., 2018), considering orography, *tas*, and *vpd* as driving factors. Ground heat flux (*G*) was assumed to correspond to 10 % of $R_n$ (Allen et al., 1998; Singer et al., 2021) and storage fluxes (*S*) were assumed to sum to zero.

In addition to general environmental conditions, information on aerodynamic and potential surface conductance were needed to calculate *pet* with the Penman-Monteith equation, and these metrics depend on the property of the surface considered. We estimated conductances for a reference crop of 12 cm height, using the simplified relationships provided by (Allen et al., 1998). $Ga$ was estimated as $\frac{w_{2*}}{208}$, with $w_{2*}$ being wind speed two metres above roughness length [m s$^{-1}$]. We derived $w_{2*}$ from our monthly estimates of *sfcWind* (which are estimated 10 m above the surface) in the following way:

$$w_{2*} = sfcWind \times \frac{ln\left(\frac{z_0+2}{z_0}\right)}{ln\left(\frac{10}{z_0}\right)}, \tag{16}$$

where $z_0$ is roughness length (see subsection Input data). $Gs_{pot}$ was calculated assuming a constant surface resistance of 70 s m$^{-1}$ (Allen et al., 1998) and considering local *tas* and pressure (using the bigleaf function 'ms.to.mol'). We calculated *pet* monthly from 1979 to 2019. For the period 1981-2010, we derived monthly climatologies of *pet* and climatological means, annual ranges and extrema. All *pet* data are reported as kg m$^{-2}$ month$^{-1}$.

### 2.2.4 Fourth-order climate layers

**Climate moisture index (*cmi*)**

Climate moisture index (*cmi*) is the difference between precipitation and potential evapotranspiration (Hogg, 1997). *cmi* informs about the moisture regime and has been related to biome boundaries, and drought impact on tree health and regeneration (Hogg et al., 2017; Hogg, 1997). We calculated *cmi* for each month of the period 1980-2018, using the CHELSA *pr* layers and the *pet* layers generated in this study. For the period 1981-2010, we derived monthly climatologies of *cmi* and climatological means, annual ranges and extrema. All *cmi* data are reported as kg m$^{-2}$ month$^{-1}$.

### 2.2.5 Fifth-order climate layers

**Site water balance (swb)**

Site water balance (*swb*) is an estimate of the water available to plants during a year, which considers soil parameters in addition to climate variables. *swb* has been shown to tightly correlate with functional plant traits such as leaf area (Grier and Running, 1977; Gholz, 1982), and it is considered one of the main determinants of plant distribution (Neilson, 1995; Woodward, 1987). We used an approach similar to that of (Grier and Running, 1977) to calculate site water balance. From the *cmi* climatologies, we identified the start of the hydrological year, i.e., either the first month after the arid period (with negative *cmi*) or the month after the one with the lowest *cmi*. Then, monthly estimates of *cmi* are summed over the hydrological year, whereby the running sum is never allowed to exceed the available water volume of the soil (approximated here by *awc*, see Input data), and excess water is assumed to run off. When *pet* exceeds precipitation (negative *cmi*) the difference is subtracted from the water balance, which often leads to distinctly negative values over the course of a hydrological year. We calculated *swb* for each year of the period 1980-2018, i.e., choosing 1981 as the first representative year and allowing hydrological years to start in 1980 already. For the period 1981-2010, we derived climatological means. All *swb* data are reported as kg m$^{-2}$ year$^{-1}$.

### 2.3 Validation

### 2.3.1 Station data

We validated nine of the 15 climate-related variables at three levels of temporal aggregation, using global sets of station measurements. We validated primarily variables that could either be measured directly or derived readily from measurements, using three different data sources. *hurs*, *sfcWind*, *fcf*, *scd*, *gdd$_5$*, and *vpd* were validated against station measurements from the Global Surface Summary of Day (GSOD) database (Global Surface Summary of Day (GSOD), 2022), containing measurements of weather conditions of >28'000 stations globally, with a focus on the northern hemisphere. We used the R package GSODR (Sparks et al., 2017) to download and quality control daily averages from 1979 to 2020, and to calculate saturation vapour pressure, actual vapour pressure, and relative humidity from measured properties, using the improved August-Roche-Magnus approximation (Alduchov and Eskridge, 1996). For each station, we then calculated vapour pressure deficit as the difference between saturation vapour pressure and actual vapour pressure, defined frost change days as days with maximum temperature >0 °C and minimum temperature <0 °C, defined daily growing degree days as average temperature minus 5°C if the average temperature was >5 °C and 0 °C otherwise, and defined snow cover days as days with measured snow depth. To validate *clt*, we used station measurements from the HadISD (v.3.2.0.2021f) global sub-daily database (Dunn, 2019), provided by the UK Met Office Hadley Centre (https://www.metoffice.gov.uk/hadobs/hadisd/). For each station, we aggregated all 1979-to-2020 hourly, non-flagged measurements of total cloud cover to daily averages and converted the original eight-level scale to percent. Station measurements for *pet* and *cmi* were obtained from the World-wide Agroclimate Data of FAO (FAOCLIM; FAO, 2001). This agro-climatic database contains data for 28,800 stations and 14 observed and

computed agro-climatic parameters. For validation, we used monthly climatologies provided by FAOCLIM version 2, which cover the period 1961-1990, and thus only partially overlap with our 1981-2010 climatologies. For potential evapotranspiration these values were available directly, while for climate moisture index we calculated them station-wise, considering only stations that simultaneously reported potential evapotranspiration and precipitation.

We aggregated station measurements temporally to three levels. Firstly, we aggregated data on *hurs*, *clt*, *sfcWind*, *fcf*, *scd*, *gdd₅*, and *vpd* by month. For *hurs*, *clt*, *sfcWind*, and *vpd*, we calculated monthly means for each combination of station and month for which 25 or more daily averages were available. For *fcf*, *scd*, and *gdd₅*, we calculated monthly sums when 25 or more daily estimates were available (for *scd*, we thereby considered temperature measurements, as snow depth was only reported when snow was present). If estimates were missing for some days, we multiplied the sum with the inverse of the fraction of days covered. Secondly, we aggregated *hurs*, *clt*, *sfcWind*, and *vpd* to monthly climatologies. To this end, we first filtered for measurements made between 1981 and 2010 and counted for each combination of month and station, how many years were available. When data for more than 15 years existed, we calculated monthly climatological means. Finally, we calculated annual climatological means. For *hurs*, *clt*, *sfcWind*, *vpd*, *pet*, and *cmi*, we did this by station-wise averaging monthly climatologies, considering stations for which estimates were missing for no more than one month. For *fcf*, *scd*, and *gdd₅*, we first derived yearly sums from 1981 to 2010, expecting 12 monthly sums per station and year. For, *scd* we did this for all combinations of stations and years with at least one observation of snow depth per year, and we further considered combinations of stations and years with daily temperature minima consistently above °C as having zero snow cover days. Then, we calculated climatological means for stations with more than 15 yearly sums.

**2.3.2 Gridded data**

In addition to station measurements, we compared CHELSA-BIOCLIM+ variables to gridded data from station-based interpolation and from a weather research and forecasting (WRF) model simulation. Gridded data from station-based interpolations originated or were built from WorldClim v2.0 (Fick and Hijmans, 2017) and from the Global Aridity Index and Potential Evapotranspiration Database version 3 (Zomer et al., 2022), and had a global coverage and spatial resolution of 30 arcsec. We calculated annual climatologies from WorldClims' monthly wind speed and solar-radiation climatologies and from the monthly climatology of from Global-AI_PET's potential evapotranspiration. Moreover, we derived estimates for relative humidity, vapor pressure deficit, and climate moisture index. We calculated relative humidity and vapor pressure using WorldClim's vapor pressure, maximum temperature, and minimum temperature, following the procedure described in Zomer et al. (2022). For climate moisture index, we subtracted Global-AI_PET's potential evapotranspiration from WorldClim's precipitation. Derived variables were first calculated for each climatological month and then averaged to annual climatologies. Note that these climatologies are representative for the period 1970-2000 and thus only partially overlap with the CHELSA-BIOCLIM+ climatologies.

For a second comparison, we considered outputs of the High Asia Refined analysis version 1 (Maussion et al., 2011, 2014) that were generated through dynamical downscaling using the WRF model version 3.3.1 (Skamarock and Klemp, 2008).

Simulated layers have a resolution of 10 km and are representative for the period 2000-2014, which only partially overlaps with the CHELSA-BIOCLIM+ climatologies. From these simulations, we used wind speed 10 meters above the surface, downward shortwave flux at ground surface (compared to *rsds*) after converting the units. Relative humidity and vapour pressure deficit were derived from daily estimates of water vapour mixing ratio (*q*), temperature at 2 m (*tas*), and surface pressure (*p*). To this end, we first calculated saturation vapour pressure from temperature, using equation 13, and actual vapour pressure according to the formula

$$e_a = \frac{q*p}{q*(1-MW_{ratio})+MW_{ratio}}, \tag{17}$$

where $MW_{ratio}$ is the ratio of molecular weights of water vapour and dry air and equals to 0.622. Relative humidity was then calculated by dividing actual vapour pressure by saturation vapour pressure and vapour pressure deficit was calculated as the difference between saturation vapour pressure and actual vapour pressure. Daily estimates of relative humidity and vapour pressure deficit were aggregated to 2000-2014 averages. Potential evaporation was converted from Watts per square meter to kilograms per square meter per year, using a linear approximation of the temperature dependency of the energy needed to vaporize water ($\Delta Hvap$ in J kg$^{-1}$)

$$\Delta Hvap = 3.148*10^6 - 2370*tas, \tag{18}$$

whereby the 2000-2014 averages of potential evaporation and *tas* were used. Finally, climate moisture index was calculated as the difference between potential evaporation and precipitation.

### 2.3.2 Summary statistics and visualizations

We matched station measurements with CHELSA-BIOCLIM+ layers and station-based interpolations at the different levels of temporal aggregation, and calculated summary statistics. For the various combinations of variable, origin (CHELSA-BIOCLIM+ or station-based interpolation), and temporal aggregation (monthly, monthly climatology, and annual climatology), we matched station-based measurements with gridded data, and we converted variables to the same units as the CHELSA-BIOCLIM+ layers. Then, we derived the number of stations for which both measurements and corresponding gridded data existed and calculated Pearson correlation coefficients (*r*), mean absolute error (MAE), root mean squared error (RMSE), absolute bias, as well as the average across station measurements. Moreover, for annual climatologies we plotted MAE in space, and we calculated and visualized *r* for each time step, for validated time-series variables (*hurs*, *clt*, *sfcWind*, and *vpd*) and for validated monthly climatologies (*hurs*, *clt*, *sfcWind*, *vpd*, *pet*, and *cmi*).

In addition to these validation results, we present detailed visualizations of spatial and temporal patterns for each variable. We show global maps as well as fine-scale patterns for one of two selected regions. For time-series variables, we report seasonal and long-term variations for different biomes, as defined by Schultz (2005), and for projected variables we show differences

between the climatological means for 1981-2010 and 2071-2100, assuming an SSP370 pathway and considering the MPI-ESM 1-2-HR model. Finally, for the Himalaya region, we visually compare the fine-scale patterns for *hurs*, *sfcWind*, *rsds*, *vpd*, *pet*, and *cmi* between CHELSA-BIOCLIM+, station-based interpolations, and WRF outputs.

## 2.4 Output format and file organization

All downscaled layers are provided as georeferenced tiff files (GeoTIFF). GeoTIFF is a public domain metadata standard which allows georeferencing information to be embedded within a TIFF file. Identical to the CHELSA layers (Karger et al., 2017), maps are projected in World Geodetic System 1984 (EPSG 4326), and have a west extent of −180.0001388888°, a south extent of −90.0001388888°, an east extent of 179.9998611111° and a north extent of 83.9998611111°. Their resolution is 0.0083333333° (30 arcsec), resulting in raster sizes of 20'880×43'200 cells. All GeoTIFF files are saved as integers with the compression option 'deflate', and an internal scale and offset (see Technical Specifications document on the CHELSA website). In order to read offset and scale correctly the geospatial data abstraction library (GDAL, https://gdal.org) version 2.2 or higher is needed, otherwise they may have to be applied manually. All variables are time-averages either representing the periods 1981-2010, 2011-2040, 2041-2070, or 2071-2100 (in case of climatologies) or individual year-month combinations (in case of time series-data). Monthly time series range at least from 1980 to 2018, while the annual time series of *swb* ranges from 1981 to 2018. Climate variable and time period, as well as SSP and Earth system model (if applicable) are encoded in the file names.

## 2.5 Software used

For the generation and validation of the climate layers, we relied on three open-source software environments. Most raster operations, such as averaging or calculating extrema were executed with SAGA V.8.1 (Conrad et al., 2015); output GeoTIFFs were created with GDAL (https://gdal.org); and validation, visualisation, as well as complex raster operations were implemented in the R environment (R Development Core Team, 2008). R packages used, in addition to those indicated above, included sp (Pebesma and Bivand, 2005), raster (Hijmans, 2019), and magick (Ooms, 2020).

## 3 Results

### 3.1 Spatiotemporal patterns

### 3.1.1 First-order climate layers

Near-surface relative humidity (*hurs*) was highest in polar regions and - to a lesser extent - at the equator, and lowest in parts of the subtropics including northern Africa, the Arabian Peninsula and northwest Australia (Figure 2a). The seasonal variation of *hurs* was most pronounced in the far north, for example in northern Canada, but also along an east-west belt in subtropical Africa, roughly from the southern tip of the Red Sea to the Atlantic Ocean. (Figure 2b). In terms of northern-hemisphere

biomes, *hurs* was lowest in the dry tropics and subtropics, especially in May and June, and highest in the polar and subpolar zone, especially in January and February (Figure 2c). Over the past forty years, annual means of *hurs* varied in all northern-hemisphere biomes with consistent and clear trends of decreasing *hurs* (Figure 2d). In the northeastern boundary region of the Andes, *hurs* tended to be higher at the northern edge of the Andes and around the eastern mountain tops than in the eastern lowlands and on the Andean plateau (Figure 2e).

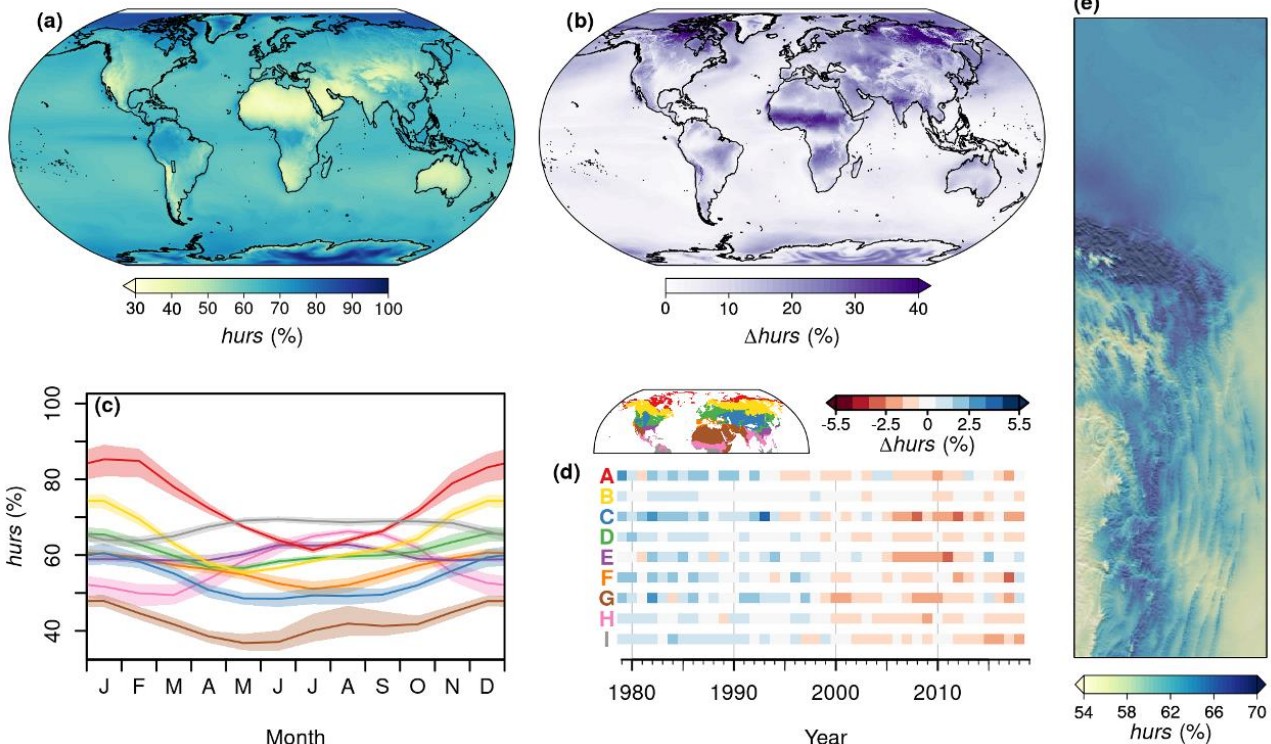


**Figure 2: Overview over the spatiotemporal distribution of near-surface relative humidity (*hurs*): a, global map of the climatological mean for the period 1981-2010; b, global map of the range (max - min) of monthly *hurs* means for the period 1981-2010; c, seasonal cycle of *hurs* in the biomes of the northern hemisphere for the period 1981-2010. Polygons indicate the range from the fortieth to the sixtieth percentile, lines indicate medians. d, temporal change of annual mean *hurs* by biome. Shown are deviations in percent of the**

**long-term (1979-2018) annual mean. Red (A) represents the polar and subpolar zone; yellow (B) represents the boreal zone; blue (C) represents dry midlatitudes; green (D) represents temperate midlatitudes; purple (E) represents subtropics with year-round**

Cloud area fraction (*clt*) was highest in polar regions and in some equatorial regions, such as Indonesia, and lowest in parts of the subtropics, including northern and South Africa and the Arabian Peninsula (Figure 3a). The seasonal variation of *clt* was most pronounced in subtropical and monsoon regions, for example on the Indian subcontinent (Figure 3b). In terms of northern-hemisphere biomes, *clt* was lowest in the dry tropics and subtropics, especially from June to August, and highest in the polar and subpolar zone, especially in May and October (Figure 3c). For the past forty years, substantial variations in annual mean

*clt* are mapped in most northern-hemisphere biomes with more (e.g., temperate midlatitudes) or less (e.g., dry tropics and subtropics) apparent negative trends (Figure 3d). In the northeastern boundary region of the Andes *clt* tended to be higher at the northern edge of the Andes and around the eastern mountain tops than in inner alpine valleys and on the Andean plateau (Figure 3e).

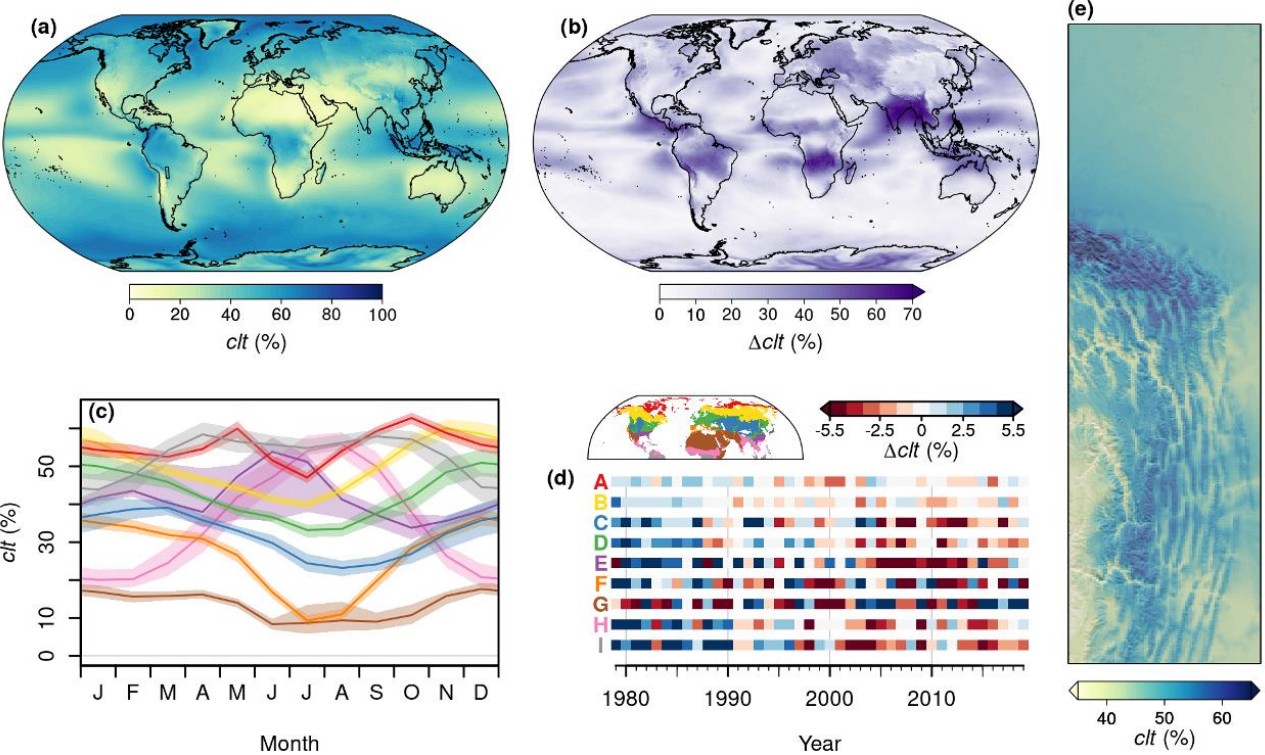

**Figure 3: Overview over the spatiotemporal distribution of cloud area fraction (*clt*): a, global map of the climatological mean for the period 1981-2010; b, global map of the range (max - min) of monthly *clt* means for the period 1981-2010; c, seasonal cycle of *clt* in the biomes of the northern hemisphere for the period 1981-2010. Polygons indicate the range from the fortieth to the sixtieth percentile, lines indicate medians. d, temporal change of annual mean *clt* by biome. Shown are deviations in percent of the long-term (1979-2019) annual mean. Red (A) represents the polar and subpolar zone; yellow (B) represents the boreal zone; blue (C) represents**

**dry midlatitudes; green (D) represents temperate midlatitudes; purple (E) represents subtropics with year-round rain; orange (F) represents subtropics with winter rain; brown (G) represents dry tropics and subtropics; pink (H) represents tropics with summer rain; and grey (I) boundary region of the Andes. e, an exemplary high-resolution map of the climatological mean of *clt* for the northeastern boundary region of the Andes. For exact location see inset in panel (a).**

Near-surface wind speed (*sfcWind*) was comparably high in the high latitudes, in coastal regions, in deserts and in mountain
systems and lowest at the equator (Figure 4a). In general, seasonal variations were relatively small, with notable exceptions of
seasonally variable *sfcWind* regions in a few, scattered regions such as Greenland and the horn of Africa (Figure 4b). In terms
of northern-hemisphere biomes, *sfcWind* was lowest in the tropics with year-round rain and highest in the polar and subpolar
zone (Figure 4c). For the past forty years, substantial variations in annual mean *sfcWind* are mapped in northern-hemisphere
biomes (Figure 4d). They show few persistent changes besides a slight increasing trend in the dry tropics and subtropics and a
slight decreasing trend in the temperate midlatitudes. In the northeastern boundary region of the Andes, *sfcWind* tended to be
highest on mountain tops, in the mideastern lowlands around the city of Santa Cruz de la Sierra, and above the lakes in the
northern lowlands of the Amazon basin (Figure 4e).

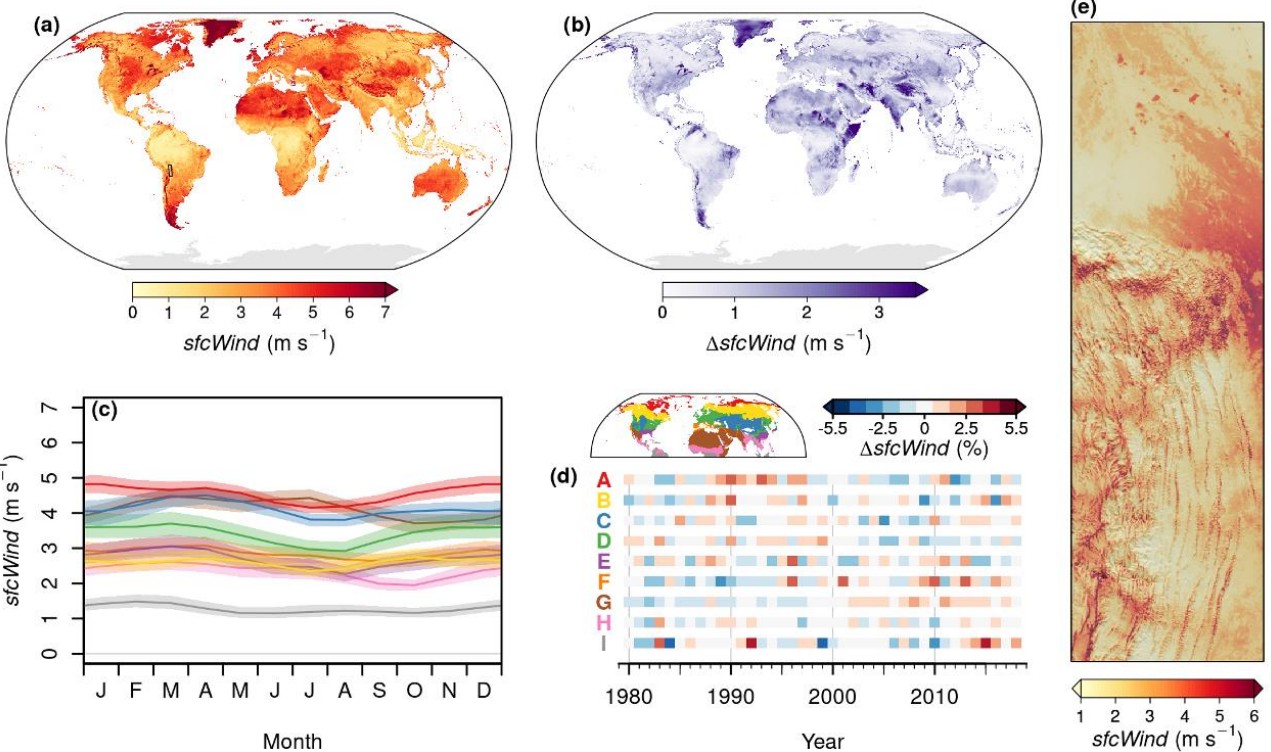

**Figure 4: Overview over the spatiotemporal distribution of near-surface wind speed (*sfcWind*): a, global map of the climatological**
**mean for the period 1981-2010; b, global map of the range (max - min) of monthly *sfcWind* means for the period 1981-2010; c,**
**seasonal cycle of *sfcWind* in the biomes of the northern hemisphere for the period 1981-2010. Polygons indicate the range from the**
**fortieth to the sixtieth percentile, lines indicate medians. d, temporal change of long-term (1980-2018) annual mean *sfcWind* by**
**biome. Shown are deviations in percent of the annual mean. Red (A) represents the polar and subpolar zone; yellow (B) represents**
**the boreal zone; blue (C) represents dry midlatitudes; green (D) represents temperate midlatitudes; purple (E) represents subtropics**
**with year-round rain; orange (F) represents subtropics with winter rain; brown (G) represents dry tropics and subtropics; pink (H)**
**represents tropics with summer rain; and grey (I) represents tropics with year-round rain. e, an exemplary high-resolution map of**
**the climatological mean of *sfcWind* for the northeastern boundary region of the Andes. For exact location see inset in panel (a).**

### 3.1.2 Second-order climate layers

Frost change frequency (*fcf*) was highest along a circumpolar belt in the temperate-to-high latitudes of the northern hemisphere
(Figure 5a), as well as in some mountain systems such as the Andes, while it was zero across most of the subtropics and tropics.
Until 2071-2100 *fcf* is expected to decrease in particular in global mountain systems and across much of the northern half of
the contiguous United States, central and eastern Europe, and southwest Asia, while increasing frost change frequencies are
expected for southeastern Canada, the Baltic countries, Belarus, Ukraine, and in Mongolia, and parts of Northern and
Northeastern China and, such as the Hengduan mountains (Figure 5b), indicating an increase in thawing events in these areas.
In the western Himalayas, *fcf* was highest at intermediate elevations, and it showed a tendency to decrease towards valley
bottoms as well as towards mountain peaks (Figure 5c).

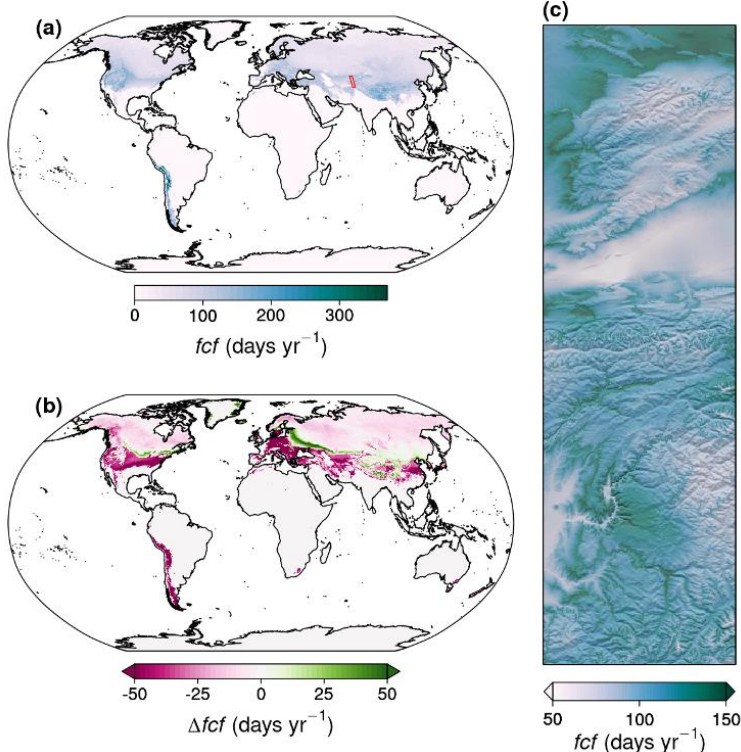

**Figure 5: Overview over the spatiotemporal distribution of frost change frequency (*fcf*): a, global map of the climatological mean of
*fcf* for the period 1981-2010; b, global map of the difference between climatological means of 2071-2100 and 1981-2010, assuming
anthropogenic emissions to follow the shared socio-economic pathway SSP370 and building on projections of the Max Planck
Institute Earth System Model (MPI-ESM 1-2-HR); c, an exemplary high-resolution map of the climatological mean for the western
edge of the Himalayas. For exact location see inset in panel (a).**

Snow cover days (*scd*) increased with latitude, with zero *scd* occurring across most of the subtropics and tropics, except for
some mountain systems, e.g., the Himalayas (Figure 6a). Until 2071-2100 *scd* are expected to decrease in all regions of the
world that currently have snow cover days, except for Greenland and Antarctica. Strongest declines are expected for the

northeastern contiguous United States and for eastern and northern Europe (Figure 6b). In the western Himalayas *scd* was positively associated with elevation (Figure 6c).

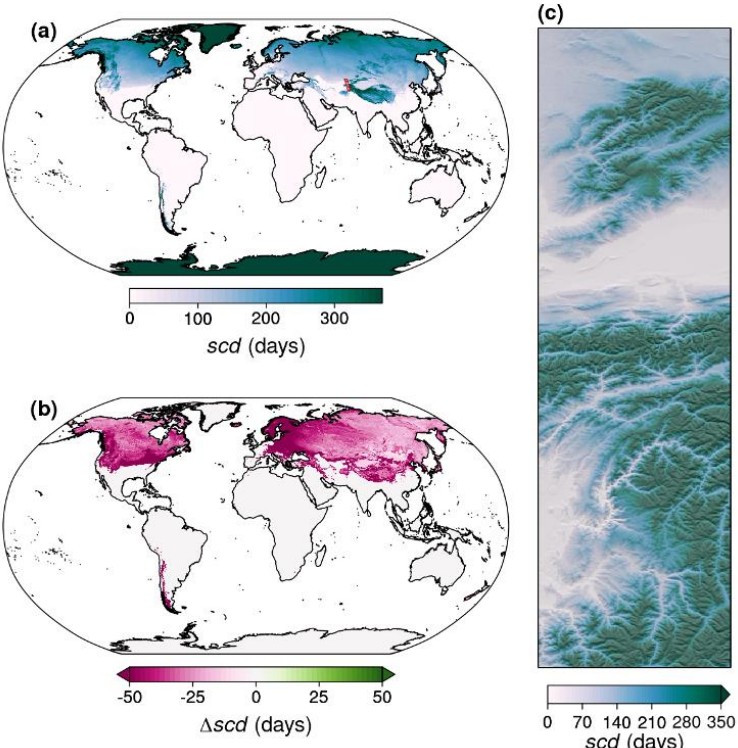

**Figure 6: Overview over the spatiotemporal distribution of snow cover days (scd): a, global map of the climatological mean of scd for the period 1981-2010; b, global map of the difference between climatological means of 2071-2100 and 1981-2010, assuming anthropogenic emissions to follow the shared socio-economic pathway SSP370 and building on projections of the Max Planck Institute Earth System Model (MPI-ESM 1-2-HR); c, an exemplary high-resolution map of the climatological mean for the western edge of the Himalayas. For exact location see inset in panel (a).**

Potential net primary productivity (*npp*) was highest in the tropics, for example in the Amazon Basin, and lowest close to the poles and in arid regions, such as northern Africa (Figure 7a). Until 2071-2100 *npp* is expected to increase across much of the northern high latitudes, in high mountain systems, and in the northwest of the Indian subcontinent. Decreasing *npp* is expected for the islands and the southern coast of the Caribbean Sea, Central America, and for the coasts of the Mediterranean Sea (Figure 7b). In the northeastern boundary region of the Andes, *npp* was highest in the northern lowlands of the Amazon Basin and lowest on the bottoms of dry inner-alpine valleys (Figure 7c).

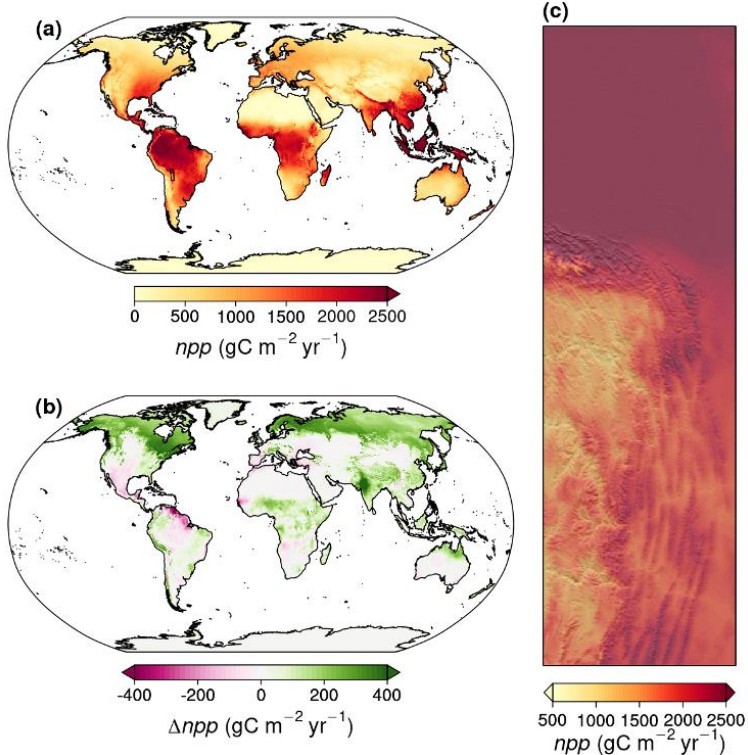


**Figure 7: Overview over the spatiotemporal distribution of net primary productivity (*npp*): a, global map of the climatological mean of *npp* for the period 1981-2010; b, global map of the difference between climatological means of 2071-2100 and 1981-2010, assuming anthropogenic emissions to follow the shared socio-economic pathway SSP370 and building on projections of the Max Planck Institute Earth System Model (MPI-ESM 1-2-HR); c, an exemplary high-resolution map of the climatological mean for the northeastern boundary region of the Andes. For exact location see inset in panel (a).**


Growing degree days with 5 °C baseline temperature (*gdd₅*) were highest in the tropics and subtropics and decreased towards the high latitudes (Figure 8a). Until 2071-2100 *gdd₅* is expected to increase in all regions of the world, except for Greenland and Antarctica. Strongest increases are expected for northern Africa and the Arabian Peninsula, Mexico, and western Australia (Figure 8b). In the northeastern boundary region of the Andes, *gdd₅* were highest in the northern lowlands of the Amazon basin

and in some inner alpine valleys, while they were lowest on high mountain peaks and the Andean plateau (Figure 8c).

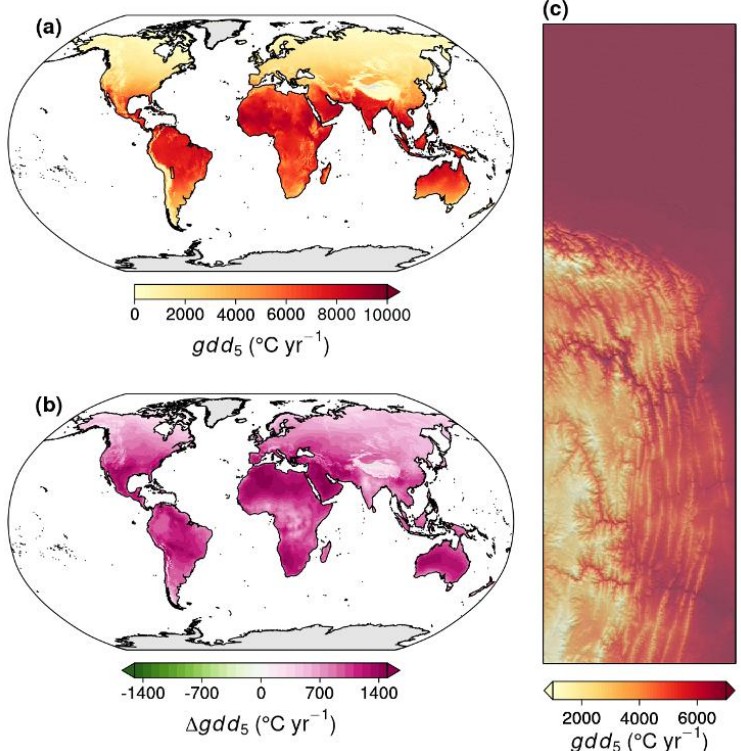

**Figure 8: Overview over the spatiotemporal distribution of growing degree days with 5 °C baseline temperature ($gdd_5$): a, global map of the climatological mean of $gdd_5$ for the period 1981-2010; b, global map of the difference between climatological means of 2071-2100 and 1981-2010, assuming anthropogenic emissions to follow the shared socio-economic pathway SSP370 and building on projections of the Max Planck Institute Earth System Model (MPI-ESM 1-2-HR); c, an exemplary high-resolution map of the climatological mean for the northeastern boundary region of the Andes. For exact location see inset in panel (a).**

The climatological mean of vapour pressure deficit (*vpd*) was highest in dry subtropical regions, for example northern Africa, the Arabian Peninsula, and Central and Western Australia. It was lowest in high mountain systems, such as the Himalayas, and polar regions (Figure 9a). The spatial patterns of seasonal variation in *vpd* were similar to those of the climatological mean (Figure 9b). In terms of northern-hemisphere biomes, *vpd* was lowest in the polar and subpolar zone, primarily from November to March, and highest in the dry tropics and subtropics, especially around June (Figure 9c). Over the past forty years annual mean *vpd* showed clearly increasing trends in all northern-hemisphere biomes (Figure 9d). In the northeastern boundary region of the Andes *vpd* showed a primary negative association with elevation, with highest *vpd* in the lowlands and in some inner alpine valleys and lowest *vpd* on mountain peaks and on the Andean plateau (Figure 9e).

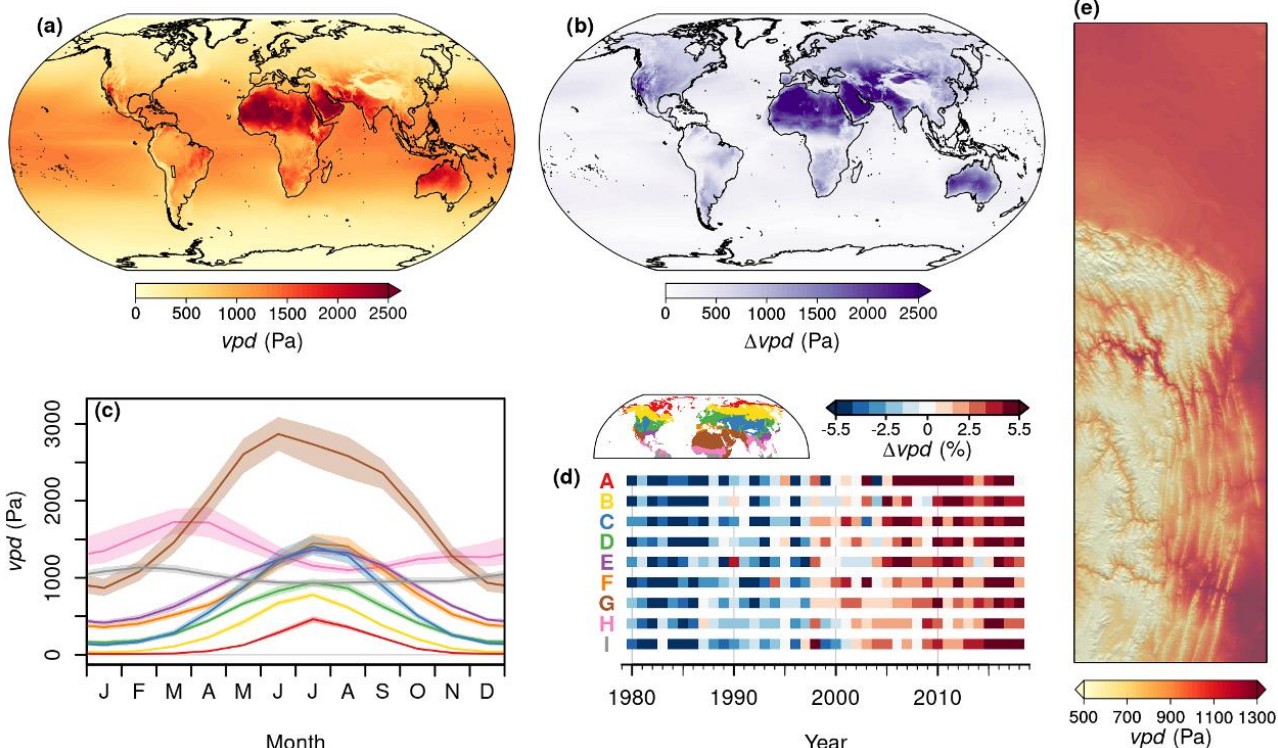

**Figure 9: Overview over the spatiotemporal distribution of vapour pressure deficit (*vpd*): a, global map of the climatological mean for the period 1981-2010; b, global map of the range (max - min) of monthly *vpd* means for the period 1981-2010; c, seasonal cycle of *vpd* in the biomes of the northern hemisphere for the period 1981-2010. Polygons indicate the range from the fortieth to the sixtieth percentile, lines indicate medians. d, temporal change of annual mean *vpd* by biome. Shown are deviations in percent of the long-term (1980-2018) annual mean. Red (A) represents the polar and subpolar zone; yellow (B) represents the boreal zone; blue (C) represents dry midlatitudes; green (D) represents temperate midlatitudes; purple (E) represents subtropics with year-round rain; orange (F) represents subtropics with winter rain; brown (G) represents dry tropics and subtropics; pink (H) represents tropics with summer rain; and grey (I) represents tropics with year-round rain. e, an exemplary high-resolution map of the climatological mean of *vpd* for the northeastern boundary region of the Andes. For exact location see inset in panel (a).**

Surface downwelling shortwave radiation (*rsds*) was highest in the subtropics and tropics, for example northern Africa and the Arabian Peninsula, and decreased towards higher latitudes (Figure 10a). The seasonal variation in *rsds* showed approximately opposite patterns, with lowest seasonal variations in the tropics, and highest variations in Antarctica and Greenland (Figure 10b). In terms of northern-hemisphere biomes, *rsds* was lowest in the polar and subpolar zone, from November to January, and highest in the dry tropics and subtropics, especially around June (Figure 10c). Over the past forty years, annual mean *rsds* showed variable trends across northern-hemisphere biomes: in several biomes, for example in the tropics with year-round rain and in particular in the subtropics with year-round rain, *rsds* tended to increase (Figure 10d) whereas in the polar and subpolar zone *rsds* tended to decrease. In the northeastern boundary region of the Andes *rsds* tended to be highest on the Andean plateau and high-elevation mountain peaks and lowest on the northern edge of the Andes on the western slopes on the western edge of the Andes (Figure 10e).

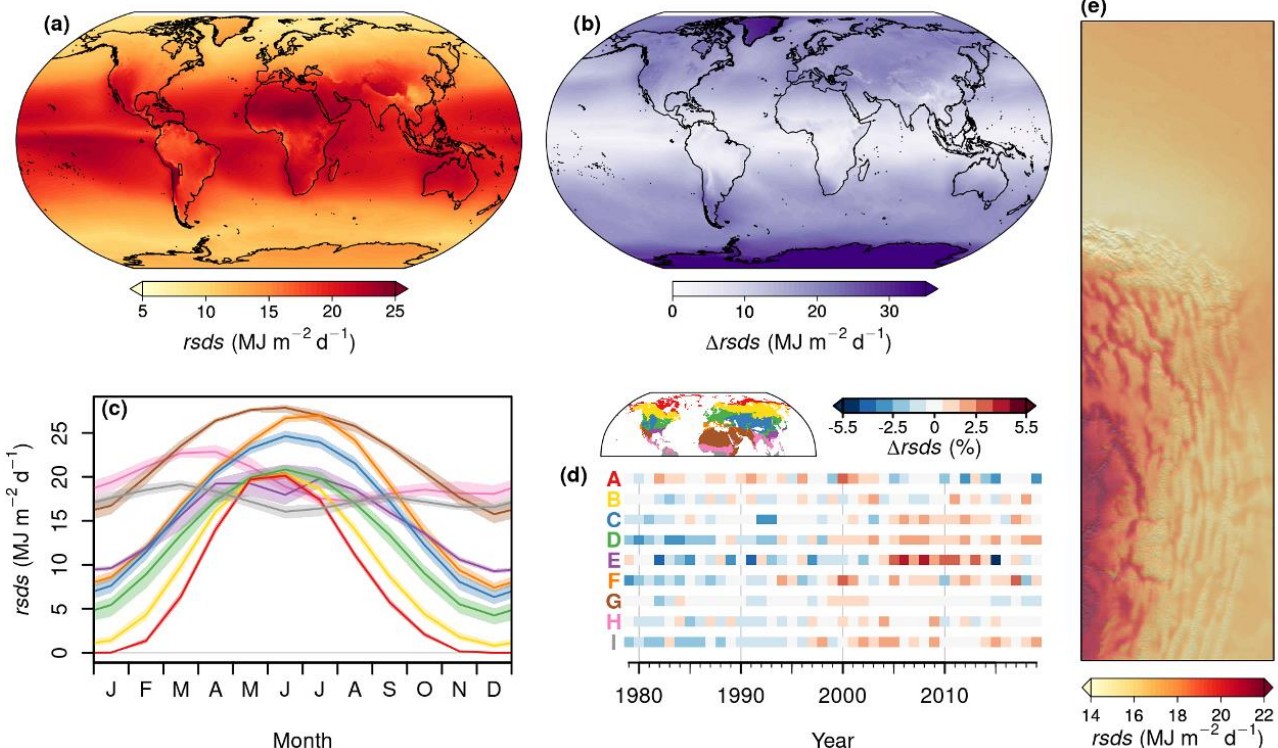

**Figure 10: Overview over the spatiotemporal distribution of surface downwelling shortwave (*rsds*): a, global map of the climatological mean for the period 1981-2010; b, global map of the range (max - min) of monthly *rsds* means for the period 1981-2010; c, seasonal cycle of *rsds* in the biomes of the northern hemisphere for the period 1981-2010. Polygons indicate the range from the fortieth to the sixtieth percentile, lines indicate medians. d, temporal change of annual mean *rsds* by biome. Shown are deviations in percent of the long-term (1979-2019) annual mean. Red (A) represents the polar and subpolar zone; yellow (B) represents the boreal zone; blue (C) represents dry midlatitudes; green (D) represents temperate midlatitudes; purple (E) represents subtropics with year-round rain; orange (F) represents subtropics with winter rain; brown (G) represents dry tropics and subtropics; pink (H) represents tropics with summer rain; and grey (I) represents tropics with year-round rain. e, an exemplary high-resolution map of the climatological mean of *rsds* for the northeastern boundary region of the Andes. For exact location see inset in panel (a).**

### 3.1.3 Third-order climate layers

Growing season length (*gsl*) was highest in the tropics, where it typically covered the entire year, and lowest in polar areas, in particular in Greenland and Antarctica, in arid areas, e.g., northern Africa, and in high mountain systems such as the Himalayas, the Rockies or the high Andes (Figure 11a). In the western Himalayas *gsl* was negatively associated with elevation (Figure 11b). Until 2071-2100 *gsl* is expected to increase across most of the temperate-to-high latitudes of the northern hemisphere and in the greater Himalaya region, but also in parts of northern Australia and central-to-eastern Africa, such as Kenya and Ethiopia. Declining growing season lengths are expected for Mexico and the southwestern US, across much of tropical South America, Spain, Morocco, and southern Australia (Figure 11c).

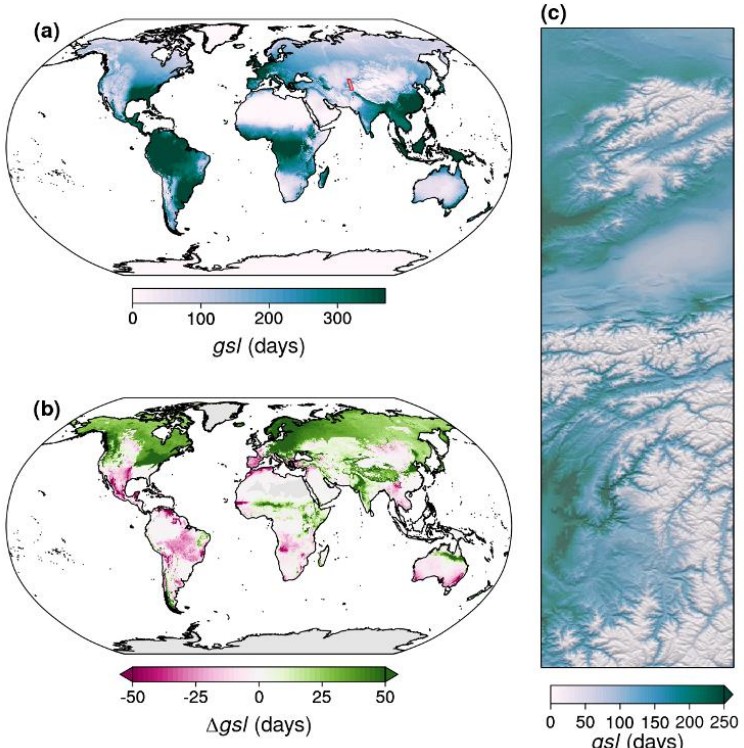

**Figure 11: Overview over the spatiotemporal distribution of growing season length (*gsl*): a, global map of the climatological mean of *gsl* for the period 1981-2010; b, global map of the difference between climatological means of 2071-2100 and 1981-2010, assuming anthropogenic emissions to follow the shared socio-economic pathway SSP370 and building on projections of the Max Planck Institute Earth System Model (MPI-ESM 1-2-HR); c, an exemplary high-resolution map of the climatological mean for the western edge of the Himalayas. For exact location see inset in panel (a).**

Growing season precipitation (*gsp*) was highest in the tropics and in the Monsoon region of Southern China, and comparably low in desert regions around the globe and in the higher latitudes, except for some coastal areas such as western North America (Figure 12a). Until 2071-2100 *gsp* is expected to increase along the coasts of western and eastern North America, across most of Eurasia, in Oceania and in northern Australia. Decreases are expected in particular in central and tropical America in the Mediterranean region, in western Africa, and in southern Australia (Figure 12b). In the northeastern boundary region of the Andes, *gsp* was highest in the northern lowlands of the Amazon Basin and in particular at the northern edge of the Andes, while it was lowest on the Andean Plateau (Figure 12c).

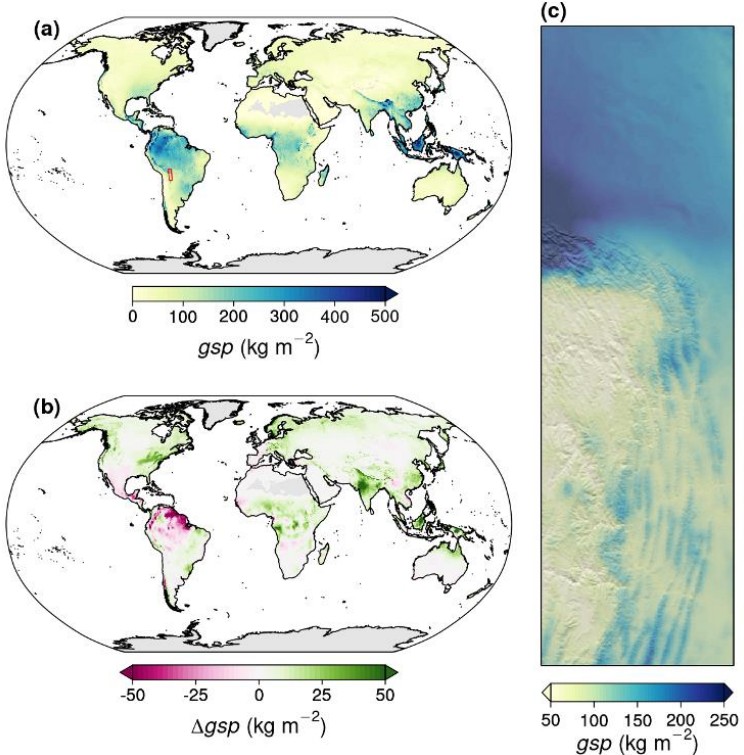

**Figure 12: Overview over the spatiotemporal distribution of growing season precipitation (*gsp*): a, global map of the climatological mean of *gsp* for the period 1981-2010; b, global map of the difference between climatological means of 2071-2100 and 1981-2010, assuming anthropogenic emissions to follow the shared socio-economic pathway SSP370 and building on projections of the Max Planck Institute Earth System Model (MPI-ESM 1-2-HR); c, an exemplary high-resolution map of the climatological mean for the northeastern boundary region of the Andes. For exact location see inset in panel (a).**

Growing season temperature (*gst*) was highest in the tropics and subtropics and decreased towards the high latitudes (Figure 13a). Until 2071-2100 *gst* is expected to increase in almost all regions of the world with growing seasons, with steepest increases for example in Mauritania. Decreasing growing season temperatures are expected, for example, from southern Sweden, over southern Ukraine to Kazakhstan (Figure 13b). In the northeastern boundary region of the Andes, *gst* was highest in the lowlands and in some inner-alpine valleys, while it was lowest on high-elevation mountain peaks and on the Andean Plateau (Figure 13c).

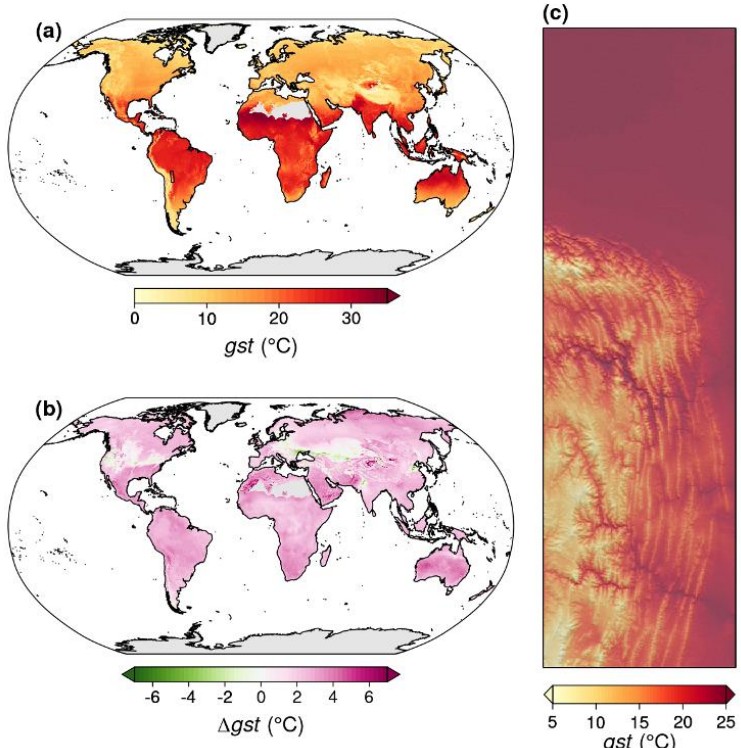

**Figure 13: Overview over the spatiotemporal distribution of growing season temperature (*gst*): a, global map of the climatological mean of *gst* for the period 1981-2010; b, global map of the difference between climatological means of 2071-2100 and 1981-2010, assuming anthropogenic emissions to follow the shared socio-economic pathway SSP370 and building on projections of the Max Planck Institute Earth System Model (MPI-ESM 1-2-HR); c, an exemplary high-resolution map of the climatological mean for the northeastern boundary region of the Andes. For exact location see inset in panel (a).**

Potential evapotranspiration (*pet*) was highest in the subtropics, such as northern Africa, and decreased towards higher latitudes, and - to a lesser extent - towards the tropics (Figure 14a). The seasonal variation of *pet* was also highest in the subtropics, but its minimum was in the tropics, and in the polar region it was intermediate (Figure 14b). In terms of northern-hemisphere biomes, *pet* was lowest in the polar and subpolar zone, from December to February, and highest in the dry tropics and subtropics, especially from May to July (Figure 14c). For the past forty years, an increasing trend of annual mean *pet* is mapped in all northern-hemisphere biomes (Figure 14d). In the northeastern boundary region of the Andes *pet* showed negative association with elevation, with lowest *pet* on high-elevation mountain peaks and on the Andean plateau and highest values in some inner alpine valleys and in the mideastern lowlands around the city of Santa Cruz. However, *pet* was also relatively low in the lowlands at the northern edge of the Andes, where *clt* and *hurs* were high and *sfcWind* and *rsds* were low (Figure 14e).

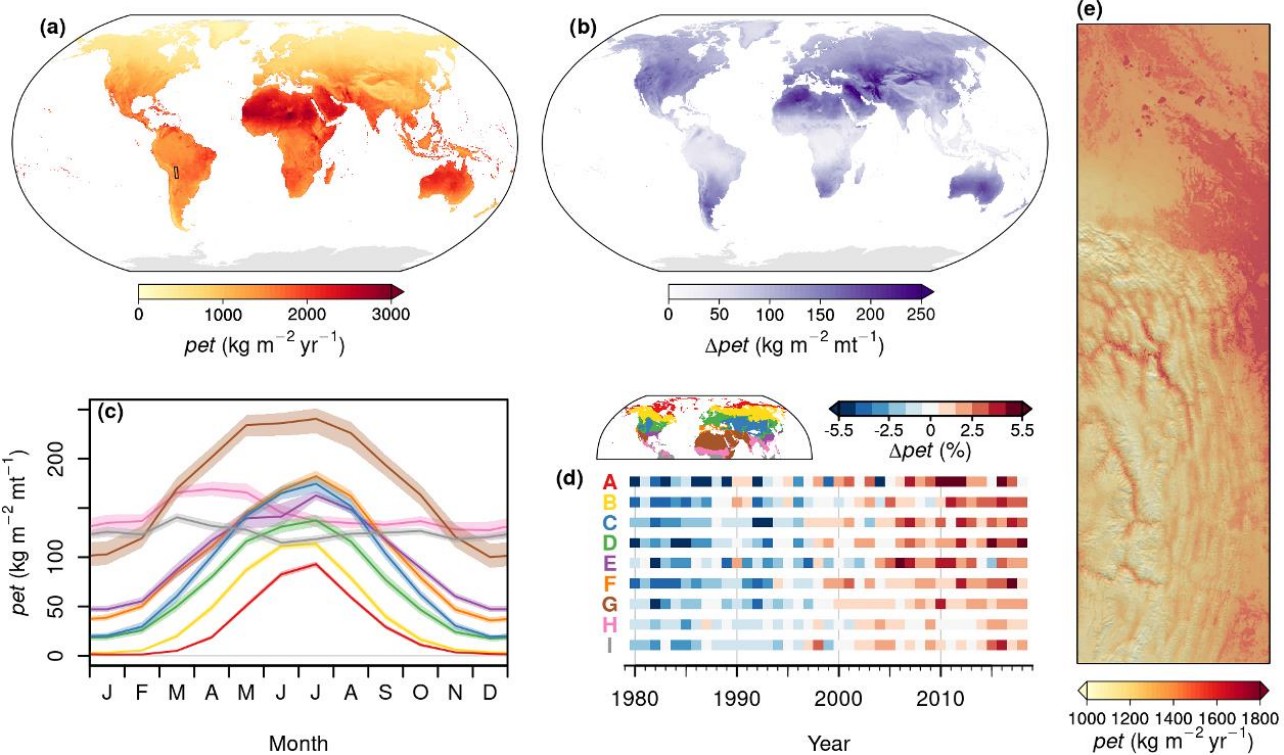

**Figure 14: Overview over the spatiotemporal distribution of potential evapotranspiration (*pet*): a, global map of the climatological mean for the period 1981-2010; b, global map of the range (max - min) of monthly *pet* means for the period 1981-2010; c, seasonal cycle of *pet* in the biomes of the northern hemisphere for the period 1981-2010. Polygons indicate the range from the fortieth to the sixtieth percentile, lines indicate medians. d, temporal change of annual mean *pet* by biome. Shown are deviations in percent of the long-term (1980-2018) annual mean. Red (A) represents the polar and subpolar zone; yellow (B) represents the boreal zone; blue (C) represents dry midlatitudes; green (D) represents temperate midlatitudes; purple (E) represents subtropics with year-round rain; orange (F) represents subtropics with winter rain; brown (G) represents dry tropics and subtropics; pink (H) represents tropics with summer rain; and grey (I) represents tropics with year-round rain. e, an exemplary high-resolution map of the climatological mean of *pet* for the northeastern boundary region of the Andes. For exact location see inset in panel (a).**

### 3.1.4 Fourth-order climate layers

Climate moisture index (*cmi*) was highest in parts of the tropics and in some mountain systems, especially in those located close to the coasts, and lowest in northern Africa and the Arabian Peninsula (Figure 15a). The seasonal variation in *cmi* was highest in the tropics and subtropics and in some coastal mountain systems such as the Pacific Northwest of North America, while in high-latitude lowlands variation was comparably low (Figure 15b). In terms of northern-hemisphere biomes, *cmi* was lowest in the dry tropics and subtropics, from May to July, and highest in the tropics with year-round rain, especially in May and June (Figure 15c). For the past forty years, substantial variations in annual mean *cmi* were observed in northern-hemisphere biomes, without clear temporal trends (Figure 15d). However, *cmi* showed a tendency to decrease in the dry tropics and subtropics. In the northeastern boundary region of the Andes *cmi* was mostly negative, in particular in inner alpine valleys, although at the northern edge of the Andes and in the lowlands of the Amazon basin *cmi* was mostly positive (Figure 15e).

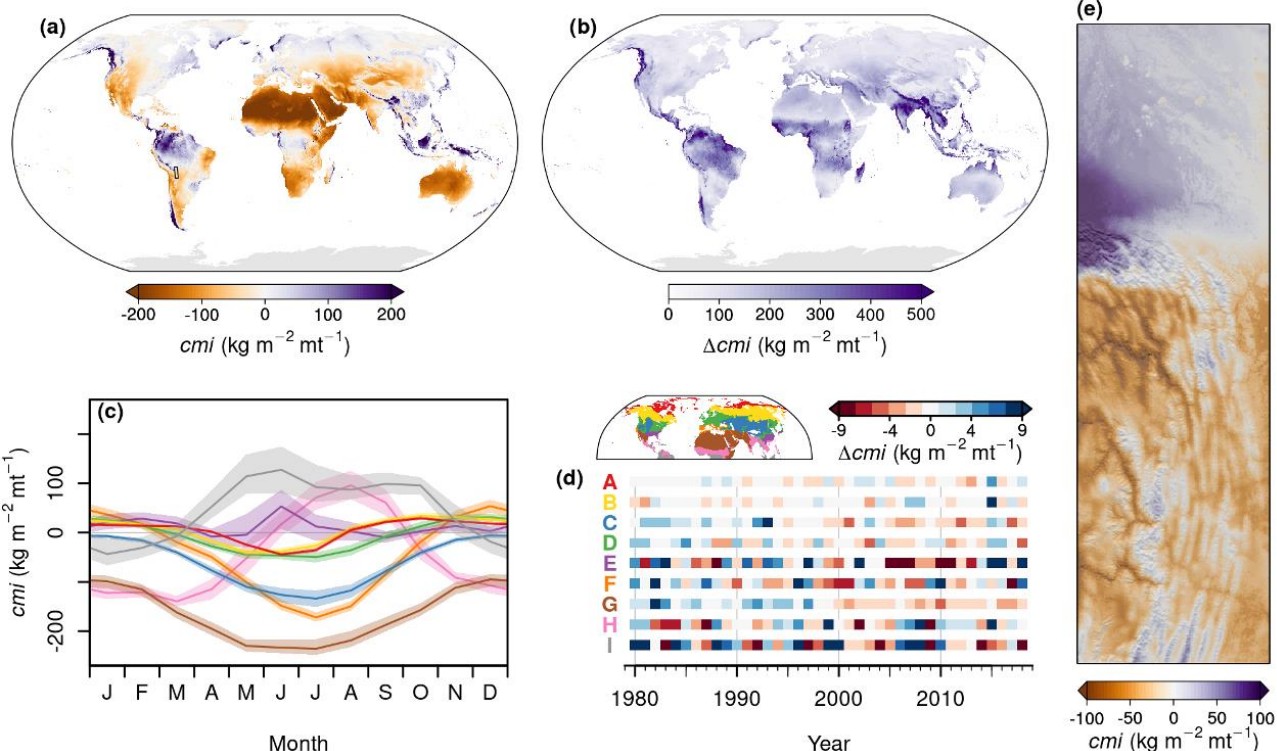

**Figure 15: Overview over the spatiotemporal distribution of climate moisture index (*cmi*): a, global map of the climatological mean for the period 1981-2010; b, global map of the range (max-min) of monthly *cmi* means for the period 1981-2010; c, seasonal cycle of *cmi* in the biomes of the northern hemisphere for the period 1981-2010. Polygons indicate the range from the fortieth to the sixtieth percentile, lines indicate medians. d, temporal change of annual mean *cmi* by biome. Shown are deviations in percent of the long-term (1980-2018) annual mean. Red (A) represents the polar and subpolar zone; yellow (B) represents the boreal zone; blue (C) represents dry midlatitudes; green (D) represents temperate midlatitudes; purple (E) represents subtropics with winter rain; orange (F) represents subtropics with year-round rain; brown (G) represents dry tropics and subtropics; pink (H) represents tropics with summer rain; and grey (I) represents tropics with year-round rain. e, an exemplary high-resolution map of the climatological mean of *cmi* for the northeastern boundary region of the Andes. For exact location see inset in panel (a).**

### 3.1.5 Fifth-order climate layers

Site water balance (*swb*) was typically neutral to positive in the tropics and in temperate-to-high latitudes while it was mostly negative elsewhere, most distinctly so in northern Africa and the Arabian Peninsula (Figure 16a). For the past forty years, substantial variations in annual mean *swb* are mapped in northern-hemisphere biomes, mostly without clear temporal trends (Figure 16b). However, *swb* did show a tendency to decrease in the dry tropics and subtropics. In the northeastern boundary region of the Andes and the surrounding lowlands, *swb* was mostly negative, in particular in inner alpine valleys, while it was slightly positive close to the northern edge of the Andes (Figure 16c).

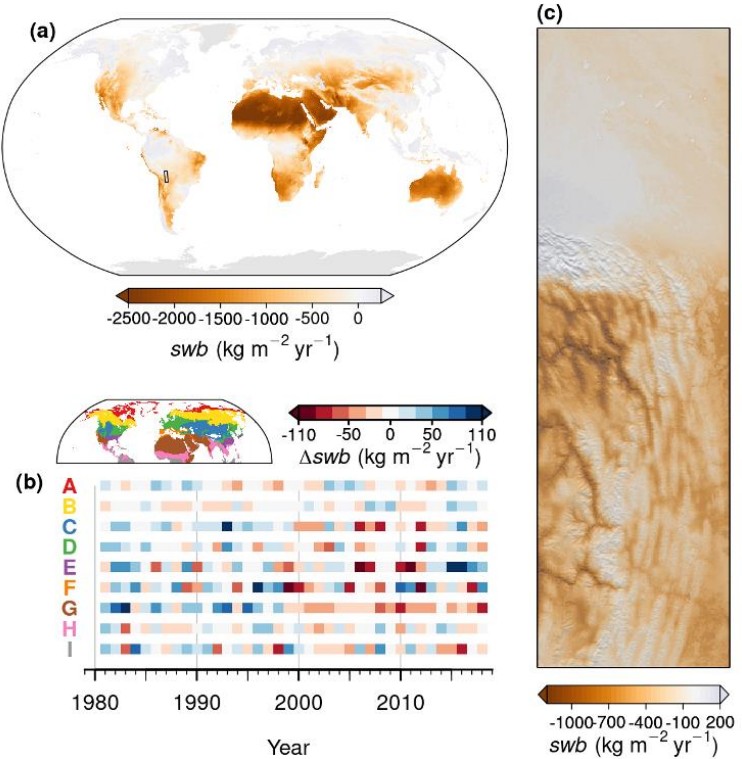

**Figure 16: Overview over the spatiotemporal distribution of site water balance (*swb*): a, global map of the climatological mean of *swb* for the period 1981-2010; b, temporal change of annual mean *swb* by biome. Shown are deviations in percent of the long-term (1980-2018) annual mean. Red (A) represents the polar and subpolar zone; yellow (B) represents the boreal zone; blue (C) represents dry midlatitudes; green (D) represents temperate midlatitudes; purple (E) represents subtropics with winter rain; orange (F) represents subtropics with year-round rain; brown (G) represents dry tropics and subtropics; pink (H) represents tropics with summer rain; and grey (I) represents tropics with year-round rain. c, an exemplary high-resolution map of the climatological mean of *swb* for the northeastern boundary region of the Andes. For exact location see inset in panel (a).**

### 3.2. Validation and comparisons

### 3.2.1 Station data

CHELSA-BIOCLIM+ layers showed a good fit with the station measurements, especially in the case of *gdd₅*, *vpd*, *hurs*, *fcf*, and *cmi* (Table 2). Pearson correlation coefficients (*r*) were high (*r* > 0.85), across all temporal aggregations evaluated, for *scd*, *gdd₅*, *vpd*, and *cmi*, and at least reasonably high (*r* > 0.80) for *hurs* and *fcf*. For *sfcWind*, correlations were lowest, yet still acceptable, with $r \geq 0.74$. For most evaluated variables, *r* was similar when evaluated for monthly and annual climatologies, with highest differences found for *pet* (*r* equalled 0.79 and 0.87 for annual and monthly climatologies, respectively). When estimated from monthly match-ups, *r* was generally lower. Over the evaluated stations, biases for annual climatologies ranged between ± 2-to-25% of the station means, except for *clt* (on average 18% too low, in absolute terms), and in particular *scd* (on average 38 days too high). MAE and RMSE were rather low for variables with comparably high *r* and low bias, such as *hurs*

(MAE of 10.81% for annual climatologies), and comparably high especially for variables with high bias, such as *scd* (MAE of 42.78 days).

Compared to station-based interpolations, CHELSA-BIOCLIM+ variables showed similar or higher performance for *hurs*, *vpd*, *pet*, and *cmi*, and somewhat lower performance for *sfcWind* (Table 2). For annual climatologies of *sfcWind*, CHELSA-BIOCLIM+ grids showed lower correlation (*r* equalled 0.77 compared to 0.84 for station-based interpolations) and higher error (MAE equalled 0.72 compared to 0.53 for station-based interpolations). For *hurs*, *vpd*, *pet*, and *cmi*, on the other hand, MAE estimates for CHELSA-BIOCLIM+ layers were lower (-17.72%, -338.17 Pa, -11.86 kg m$^{-2}$ mt$^{-1}$, and -8.73 kg m$^{-2}$ mt$^{-1}$,

respectively), and *r* was similar or higher (+0.24, +0.06, +0.13, and -0.01, respectively) compared to corresponding metrics for station-based interpolations.

**Table 2: Validation results for nine evaluated variables. *r* represents Pearson correlation coefficient; MAE stands for mean absolute error; RMSE stands for root mean squared error; Mean indicates the averages of station measurements; Bias represents the average**
**difference between gridded estimates and station measurements. Units are as reported in the methods.**

| Variable | Aggregation | Origin | Validation data | *r* | MAE | RMSE | Mean | Bias | Stations |
|---|---|---|---|---|---|---|---|---|---|
| *hurs* | Climat. mean[†] | This study | GSOD | 0.90 | 10.81 | 11.84 | 69.62 | -10.00 | 4412 |
| *hurs* | Climat. month | This study | GSOD | 0.88 | 11.45 | 12.71 | 69.68 | -10.03 | 5702 |
| *hurs* | Monthly | This study | GSOD | 0.84 | 11.91 | 13.45 | 69.72 | -10.03 | 17'316 |
| *hurs* | Climat.[†] mean | Station-based[‡] | GSOD | 0.66 | 28.53 | 30.44 | 69.11 | 28.51 | 4143 |
| *clt* | Climat.[†] mean | This study | HadISD | 0.87 | 18.07 | 19.21 | 55.76 | -18.03 | 5095 |
| *clt* | Climat.[†] month | This study | HadISD | 0.86 | 18.12 | 19.70 | 55.87 | -17.94 | 5989 |
| *clt* | Monthly | This study | HadISD | 0.79 | 18.01 | 20.66 | 55.09 | -17.00 | 8323 |
| *sfcWind* | Climat.[†] mean | This study | GSOD | 0.77 | 0.72 | 0.94 | 3.38 | 0.05 | 4482 |
| *sfcWind* | Climat.[†] month | This study | GSOD | 0.78 | 0.76 | 1.00 | 3.37 | 0.06 | 5782 |
| *sfcWind* | Monthly | This study | GSOD | 0.74 | 0.87 | 1.17 | 3.33 | 0.14 | 17'385 |
| *sfcWind* | Climat.[†] mean | Station-based[‡] | GSOD | 0.84 | 0.53 | 0.74 | 3.28 | -0.01 | 4223 |
| *fcf* | Climat.[†] mean | This study | GSOD | 0.82 | 19.76 | 27.26 | 50.99 | -4.59 | 4101 |
| *scd* | Climat.[†] mean | This study | GSOD | 0.91 | 42.78 | 62.01 | 50.69 | 38.09 | 2283 |
| *gdd₅* | Climat.[†] mean | This study | GSOD | 0.99 | 159.14 | 239.16 | 3358.40 | -67.00 | 4085 |
| *vpd* | Climat.[†] mean | This study | GSOD | 0.91 | 177.01 | 219.18 | 582.36 | 135.59 | 4143 |
| *vpd* | Climat.[†] month | This study | GSOD | 0.93 | 194.56 | 255.94 | 599.04 | 194.56 | 5702 |
| *vpd* | Monthly | This study | GSOD | 0.92 | 205.82 | 278.16 | 598.43 | 134.80 | 17'316 |

| *vpd* | Climat.[†] mean | Station-based[‡] | GSOD | 0.85 | 515.18 | 579.60 | 595.40 | -515.04 | 4143 |
| *pet* | Climat.[†] mean | This study | FAOCLIM | 0.79 | 19.84 | 24.12 | 120.77 | 6.18 | 4247 |
| *pet* | Climat.[†] month | This study | FAOCLIM | 0.87 | 21.70 | 27.03 | 120.77 | 6.18 | 4206 |
| *pet* | Climat.[†] mean | Station-based[‡] | FAOCLIM | 0.66 | 31.70 | 37.04 | 121.16 | 20.30 | 4050 |
| *cmi* | Climat.[†] mean | This sutdy | FAOCLIM | 0.88 | 27.35 | 40.50 | -27.78 | -2.94 | 4207 |
| *cmi* | Climat.[†] month | This study | FAOCLIM | 0.91 | 34.24 | 55.07 | -21.78 | -2.94 | 4166 |
| *cmi* | Climat.[†] mean | Station-based[‡] | FAOCLIM | 0.89 | 36.08 | 45.71 | -23.74 | -21.01 | 4011 |

[†] climatology for the period 1981-2010; [‡] derived from WorldClim v2.0 and the Global Aridity Index and Potential Evapotranspiration Database version 3 (Fick and Hijmans, 2017; Zomer et al., 2022)

Mean absolute error of CHELSA-BIOCLIM+ variables showed variable distributions in space. For *hurs*, MAE was high in Europe and Southeast Asia, and comparably low in western North America and temperate-to-boreal Asia (Figure 17a). For *fcf*, MAE was particularly high for an area extending from southeast Europe eastwards into central Asia, while it was low for the subtropics and tropics (Figure 17d). For *gdd₅*, elevated MAE was mainly found in the subtropics and tropics, especially in northern Mexico and at the northern edge of the Andes (Figure 17f). For *clt*, *sfcWind*, *pet*, and *cmi*, the patterns were roughly uniform, although some regions showed a somewhat elevated error, for example Niger for *clt* or Mongolia for *pet* and *cmi* (Figure 17b,c,h,i). For *scd* and *vpd*, which had a comparably high bias (Table 2), MAE showed a latitudinal pattern that was roughly proportional to the primary pattern of the variable (Figure 17e,g).

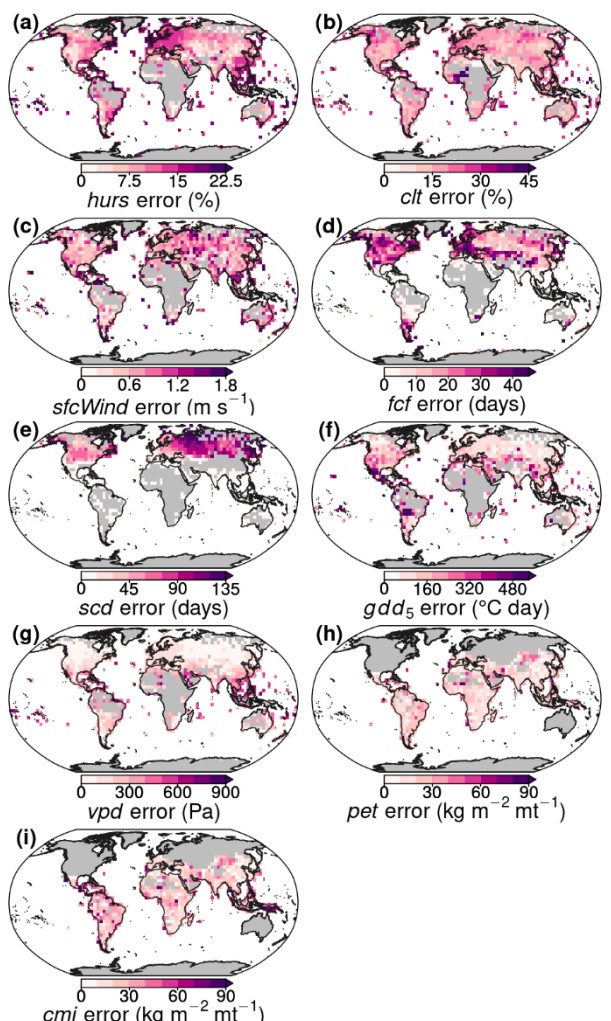

**Figure 17: Spatial distribution of validation errors: global maps of mean absolute errors between 1981-2010 climatological means of nine CHELSA-BIOCLIM+ variables and corresponding averages of station estimates. Note that for *pet* (h) and *cmi* (i) station data is representative for the period 1961-1990, and thus time periods only partially overlap.**

Pearson correlation coefficients between station measurements and evaluated CHELSA-BIOCLIM+ variables varied with season and between years. Seasonal variations were particularly pronounced for *hurs*, where *r* was below 0.8 for January and December, and above 0.9 from April to October (Figure 18a). For *sfcWind* and *pet*, a clear seasonal signal in *r* also existed, but highest correlations ($r > 0.8$ and $r > 0.9$, respectively) were found from November to February, and lowest correlations in July, for *sfcWind* ($r = 0.72$), and August, for *pet* ($r = 0.78$). For *vpd* and *cmi*, on the other hand, seasonal variations were

comparably small. Inter-annual variations in Pearson correlation coefficients were pronounced for *clt* and *sfcWind* while they were relatively small for *vpd* and *hurs* (Figure 18b). Apart from the seasonal variations, *r* for *clt* and *sfcWind* remained relatively stable between 1980 and 1995 (average 1980-1995 *r* was 0.74 and 0.82 for *sfcWind* and *clt*, respectively). Between 1995 and the early two-thousands, *r* declined for both variables, before it started increasing again until about 2010. After that,

*r* for both variables declined a second time until the end of the time series (average 2010-to-time series end *r* equalled 0.72

and 0.76, for *sfcWind* and *clt*, respectively).

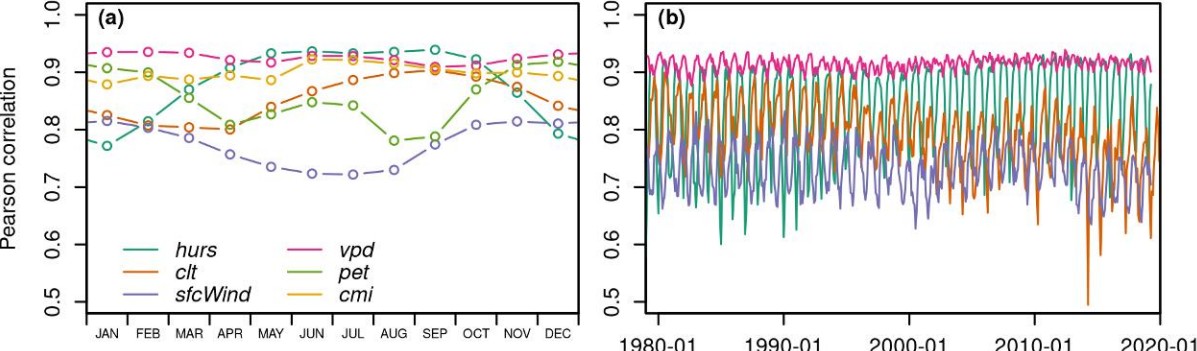

**Figure 18: Seasonal and inter-annual distribution of correspondence: a, Person correlation coefficients between CHELSA-BIOCLIM+ variables and corresponding averages of station measurements for *hurs*, *clt*, *sfcWind*, *vpd*, *pet*, and *cmi* for each 1981-2010 climatological month. Note that for *pet* (purple) and *cmi* (yellow) station data is representative for the period 1961-1990, and**
**thus time periods only partially overlap. b, Person correlation coefficients between CHELSA-BIOCLIM+ variables and corresponding averages of station measurements for *hurs*, *clt*, *sfcWind*, and *vpd* for each month in the time series. Same colors are used as in panel a.**

### 3.2.2 Gridded data

In the Himalaya region, the spatial patterns of CHELSA-BIOCLIM+ variables were generally similar to those of corresponding

layers from station-based interpolations and weather research and forecasting simulations, although some exceptions existed. The spatial patterns of *hurs* were comparably variable among products, with highest correlation between CHELSA-BIOCLIM+ and WRF ($r = 0.68$, Figure 19a-c). For *sfcWind*, the large-scale patterns were quite similar, especially between CHELSA-BIOCLIM+ and WRF, but the fine-scale structures were resolved in higher detail in the CHELSA-BIOCLIM+ layers explaining why the correlation between station-based interpolations and WRF was highest for wind speed (Figure 19d-

f). For vapour pressure deficit, the patterns between all products were very similar (Figure 19g-i). In the case of *rsds*, the CHELSA-BIOCLIM+ layer showed the most pronounced fine-scale patterns, and its large-scale patterns were similar to those in the WRF layer ($r = 0.73$ between CHELSA-BIOCLIM+ and WRF). The patterns of WorldClim's solar radiation, on the other hand, were strikingly different compared to the former two products ($r < 0$ for both comparisons, Figure 19j-l). For *pet*, large-scale patterns between the three products were generally similar, although absolute values were somewhat lower for

CHELSA-BIOCLIM+ and more fine-scale structures were visible (Figure 19m-o). In the case of *cmi* the patterns were generally similar (Figure 19m-o). Along the southern edge of the Himalayas, the large-scale patterns between CHELSA-BIOCLIM+ and WRF were somewhat more similar than those between CHELSA-BIOCLIM+ and station-based interpolations.

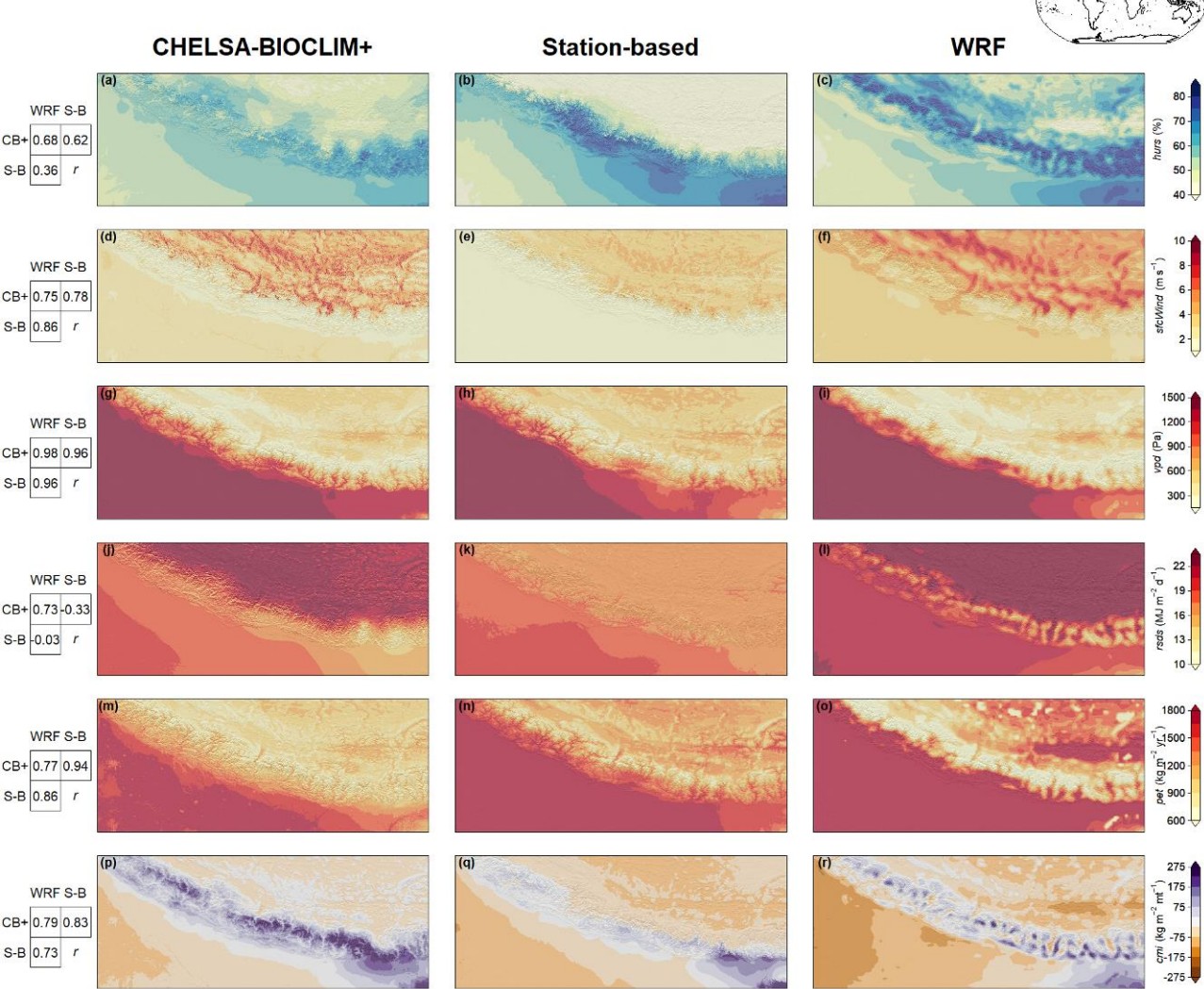

**Figure 19: Spatial patterns in a data-poor region - a comparison with existing products.** CHELSA-BIOCLIM+ variables are compared with equivalents from station-based interpolation and form a weather research and forecasting (WRF) model for *hurs* (a-c), *sfcWind* (d-f), *vpd* (g-i), *rsds* (j-l), *pet* (m-o), and *cmi* (p-r) in the Himalaya region (see inset map on the top right). On the left, for each variable pairwise Pearson correlation coefficients are shown for the mapped area, between CHELSA-BIOCLIM+ (CB+), the weather research and forecasting model (WRF), and station-based interpolations (S-B). Station-based interpolations are derived from WorldClim v2.0 and the Global Aridity Index and Potential Evapotranspiration Database version 3 (Fick and Hijmans, 2017; Zomer et al., 2022); the WRF simulation considered was the High Asia Refined analysis v1 (Maussion et al., 2014, 2011).

**4 Discussion**

Climate data at high spatiotemporal resolution for current conditions and for the decades ahead of us are crucial to fill the gaps in our understanding of climate-change impacts on the Earth system. Here, we provide a dataset of biologically meaningful, essential climate and environmental variables, combining state-of-the-art input data with a mechanistic downscaling methodology. The provided gridded layers offer unprecedented spatiotemporal resolution and high validation accuracy. Characterizing bioclimate comprehensively beyond temperature and precipitation, makes our data set particularly relevant to

study biological processes (Bojinski et al., 2014; Woodward, 1987; Neilson, 1995). The open-access dataset CHELSA-BIOCLIM+ will stimulate research on climate-change impact on physical and ecological processes.

Comprehensive information on climate beyond temperature and precipitation enables better characterisation of various Earth system processes, and biological processes in particular, where the balance between water supply and energy demand is central

(Woodward, 1987). Our time-series variables related to water availability (*hurs*, *vpd*, *pet, cmi*) and incoming solar energy (*clt*, *rsds*) matched particularly well with validation data. Moreover, they showed low error in comparison to estimates derived from station-based interpolations. They can thus provide valuable inputs to a variety of downstream analyses such as analysing the distribution of leaf area (Grier and Running, 1977; Iio et al., 2014), primary productivity (Gholz, 1982; Aguilos et al., 2021) or of plant functional type-based biomes (Neilson, 1995; Schultz, 2005). For *swb*, suitable data for direct validation are

scarce. However, given that only an additional estimate of soil water bucket size went into the calculation of this variable, we can expect that its performance is comparable to the input variables it was computed from. *rsds* was not validated here, but Karger et al. (submitted to ESSD) demonstrated that, on a daily basis, global *rsds* estimates matched very well with *in situ* measurements ($r = 0.89$). Moreover, in the Himalaya region matched the spatial pattern of *rsds* matched well with dynamically downscaled WRF outputs (Figure 19). A key strength of the CHELSA-BIOCLIM+ product therefore lies in the provision of

accurate, high-resolution, global time-series of climate-related variables describing the true availability of water and solar energy.

Combining input data from reanalysis with mechanistic downscaling approaches allows for robust estimates in particular in remote areas. So far, climate data used in macroecological analyses often relied on station-based interpolations (Bobrowski et

al., 2021). While such data may be accurate in regions that have a dense network of field stations, such as Europe and North America (Hijmans et al., 2005), they are much less reliable in remote areas, and/or in complex terrain (Karger et al., 2017). The CHELSA approach, on the other hand, uses gridded reanalysis data that account for physical mesoscale atmospheric processes and physical consistency (Hersbach et al., 2020), and further considers major orographic effects such as the shading of terrain or wind exposure (e.g., for *hurs*, *clt*, and *pet*, see methods). Given the lack of field stations in remote areas, the extent

of these improvements is likely not fully mirrored in the validation results, although our comparison among different products in the Himalaya region highlights that CHELSA-BIOCLIM+ layers generally compare well to alternative products. Moreover,

CHELSA-based estimates of temperature and precipitation, which fully or partially underlie most variables presented here, have repeatedly been shown to be better-suited than station-based interpolations for ecological modelling in the remote Himalayas (Datta et al., 2020; Suwal et al., 2018). It may therefore be expected that the CHELSA-BIOCLIM+ product is
particularly advantageous in remote areas.

Generating a comprehensive global set of high-resolution climate-related variables requires making generalising assumptions that can compromise the accuracy of some estimates. Specifically, our projected variables providing current and future estimates are sensitive to bias, since simple models were preferred to make robust projections (Levins, 1966). As highlighted
by the validation, our estimates of *scd* overestimated station-based measurements by about one month in regions with snow. These differences may arise from generating estimates of daily *tas*, *tasmin*, *tasmax*, and *pr* from monthly averages by means of spline interpolation, which results in a more gradual seasonal evolution of temperature and precipitation than observed in natural weather patterns. Moreover, for the computation of *scd*, contributing factors such as solar radiation were ignored. Similarly, the model to generate estimates for *gsl*, *gsp*, and *gst* only contained a simplistic implementation of soil water
processes (Paulsen and Körner, 2014) and the Miami model to generate estimates for *npp* ignored soil conditions and solar radiation entirely. However, the approaches used to generate projected variables were not primarily selected for their accuracy, but for their generalism (Levins, 1966) to be applicable under current *and* projected future conditions and to avoid overfitting. Despite significant advances during the past years (Kawamiya et al., 2020) Earth system models are still not capable of fully resolving mesoscale weather processes and thus they are primarily suited to study long-term changes in climate rather than
possible weather patterns (Held et al., 2019; Yukimoto et al., 2019; Gutjahr et al., 2019; Boucher et al., 2020). Relative to our time-series variables, our projected variables may therefore not offer the same high accuracy for the recent past, but they approximate climate-change impact on fundamental biological and ecological quantities, such as potential net primary productivity, and make them directly comparable for a variety of possible future conditions, building on the most accurate global prognoses that are currently available (Eyring et al., 2016).


The validation also revealed inaccuracies for the time-series variable *sfcWind*. Although in the remote Himalaya region the *sfcWind* grids of CHELSA-BIOCLIM+ compared well to dynamically downscaled *sfcWind* from WRF, the correspondence to station measurements was weaker than for gridded data from station-based interpolations. Moreover, the monthly Pearson correlation coefficients between grids and station measurements declined somewhat for recent years. A reason for the higher
correlation of the station-based interpolations might be that the station measurements we used here for validation largely overlap with their input data (Fick and Hijmans, 2017), and thus they are expected to perform well in our validation. Downscaling wind fields form ERA5, on the other hand, is challenging, as wind inherently contains a high variance that can be reconstructed to a limited degree even with the most sophisticated downscaling approaches (Pryor and Hahmann, 2019). Perhaps even more importantly, wind fields from reanalysis by themselves are of limited accuracy. Global meteorological
stations indicate that wind speed has been declining from the 1980s until around 2010 and has recovered afterwards. In

reanalysis products, however, this striking pattern is hardly reproduced (Zeng et al., 2019). We will keep updating and improving the CHELSA family of climate data products, and this is especially true for *sfcWind*, should better input data become available.

In conclusion, CHELSA-BIOCLIM+ is a comprehensive spatial and temporal data set of 15 climate-related variables including both, time-series for the past forty years and future projections building on several SSPs and Earth system models. Besides the climatological statistics provided, these data may be used to compute additional summaries, for example interannual variabilities of *cmi*, which are important factors determining ecosystem structure in the dry midlatitudes, subtropics and tropics (Schultz, 2005). Moreover, the downscaling pipeline developed here opens new perspectives to develop near real-time risk

assessments when regularly updated and combined with machine learning and increasingly available global phenomenological datasets. The higher temporal resolution and the more proximal variables included in the CHELSA-BIOCLIM+ product will allow for a more detailed characterization of climate-related conditions and, in turn, a deeper understanding of their impact on key environmental processes.

**Data availability**

The CHELSA-BIOCLIM+ data set consists of 4006 single-layer GeoTIFF files, representing averages, extrema, and ranges of the 15 climate-related variables for different time points (1979 to 2100) and periods (monthly to 30-year-averages). The GeoTIFF files are stored on a S3 cloud server that can be accessed over EnviDat (https://doi.org/10.16904/envidat.332; (Brun et al., 2022), by clicking on the 'CHELSA-BIOCLIM+' box in the 'Data and resources' tab, and over www.chelsa-climate.org, by clicking on 'Version 2.1' under 'Downloads'. This file browser contains the four folders 'annual', 'daily', 'monthly', and

'climatologies' within which the CHELSA-BIOCLIM+ data are organized in the following way:

- The folder 'annual' contains the subfolder 'swb', which contains annual layers of *swb*.
- The folder 'daily' contains no data of the CHELSA-BIOCLIM+ data set.
- The folder 'monthly' contains (among folders from other data sets) the subfolders 'clt', 'cmi', 'hurs', 'pet', 'rsds', 'sfcWind', and 'vpd' which contain monthly layers for *clt*, *cmi*, *hurs*, *pet*, *rsds*, *sfcWind*, and *vpd*, respectively.

- The folder 'climatologies' contains four subfolders, '1981-2010', '2011-2040', '2041-2070', and '2071-2100', that represent the different time periods for which climatologies are representative.
  - In the subfolder '1981-2010' the sub-subfolders 'clt', 'cmi', 'hurs', 'pet', 'rsds', 'sfcWind', and 'vpd' contain 1981-2010 averages of *clt*, *cmi*, *hurs*, *pet*, *rsds*, *sfcWind*, and *vpd*, respectively, for each month. The sub-subfolder 'bio' contains (among files from other data sets) climatological means, maxima, minima and

annual ranges for *clt*, *cmi*, *hurs*, *pet*, *rsds*, *sfcWind*, and *vpd*, and climatological means for *fcf*, *gdd* (with 0 °C, 5 °C, and 10 °C baseline temperature, i.e., 'gdd0', 'gdd5', 'gdd10', respectively), *gsl*, *gsp*, *gst*, *npp*, *scd*, and *swb*.

o The subfolders '2010-2040', '2041-2070', and '2071-2100' each contain one sub-subfolder per Earth system model considered (i.e., the sub-subfolders 'GFDL-ESM4', 'IPSL-CM6A-LR', 'MPI-ESM1-2-HR', 'MRI-ESM2-0', 'UKESM1-0-LL'). Each of these combinations between period and Earth system model contains three sub-sub-subfolders representing the three SSPs (i.e., the sub-sub-subfolders 'ssp126', 'ssp370', and 'ssp585'); and each of these combinations between period, Earth system model, and SSP, contains a sub-sub-sub-subfolder 'bio' that contains (among files from other data sets) climatological means for *fcf*, *gdd* (with 0 °C, 5 °C, and 10 °C baseline temperature), *gsl*, *gsp*, *gst*, *npp*, and *scd*.

More information on naming and settings of the GeoTIFF files (grid structure, unit, scale and offset parameters) can be found in the subsection '2.4 Output format and file organization' and in the Technical Documentation PDF that can be found on https://doi.org/10.16904/envidat.332 in the 'CHELSA-BIOCLIM+ Technical Documentation' box in the 'Data and resources' tab. Monthly and annual layers of the time-series variables will occasionally be added to the CHELSA-BIOCLIM+ data set, to extend the time period covered to the most recent years.

**Author contribution**

PB, DNK, NEZ, and LP conceived the general idea of the paper. PB and DNK generated the data set. CH conducted the validation with support of PB and DNK. PB led the writing of the manuscript. All authors significantly contributed to writing and editing.

**Competing interests**

The authors declare that they have no conflict of interest.

**Acknowledgements**

DNK, LP & NEZ acknowledge funding from: the WSL internal grant exCHELSA, the 2019–2020 BiodivERsA joint call for research proposals, under the BiodivClim ERA-Net COFUND program, with the funding organisations Swiss National Science Foundation SNF (project: FeedBaCks, 193907), as well as the Swiss Data Science Project: SPEEDMIND. PB, DNK & NEZ the Swiss Data Science Project: COMECO. DNK acknowledges funding to the ERA-Net BiodivERsA - Belmont Forum, with the national funder Swiss National Science Foundation (20BD21_184131), part of the 2018 Joint call BiodivERsA-Belmont Forum call (project 'FutureWeb'), as well as the WSL internal grant ClimEx. We thank Babek Dabagchian for valuable support in data management and preparation.

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
