# Peer review of "Global climate-related predictors at kilometre resolution for the past and future"

_Earth System Science Data, 2022_

## Author Comment (AC1)

**Swiss Federal Research Institute WSL**
Eidg. Forschungsanstalt WSL
Institut fédéral de recherches WSL
Istituto federale di ricerca WSL

Editorial Support
Earth System Science Data
Copernicus Publications

Birmensdorf, October 31, 2022

[Figure]

Dr. Philipp Brun
Dynamic Macroecology Group
Phone +41-44-739 22 45, fax +41-44-739 22 15
philipp.brun@wsl.ch

Dear Editor,

Please find attached the resubmitted manuscript, "Global climate-related predictors at kilometre resolution for the past and future". We would like to thank you and the two referees for taking the time to evaluate our manuscript and for the very valuable and constructive feedback. We have carefully considered all the comments and changed the revised manuscript accordingly. In particular, we have substantially expanded the validation. We have increased the number of validated variables, report validation results for different temporal aggregations, and compare them to alternative products. The validation table (now Table 2) is now substantially enlarged, Figure 17 shows how error is distributed in space, Figure 18 shows how Pearson correlation varies over the season and between years, and Figure 19 compares spatial patterns of BIOCLIM+ variables to station-based interpolations and weather research and forecasting model outputs in the Himalaya region. Moreover, we have added a table detailing all input data used. Below, we list in detail our responses to the comments, and how we incorporated them into the manuscript. The comments of the reviewers below are shown in black, and our responses are highlighted in blue.

Thank you very much for considering this manuscript, and we look forward to your reply.

Yours sincerely,

Philipp Brun
(on behalf of the authors)

**Detailed response to comments of Referee 1**

*R1.1:*
This study produces high spatial resolution climate datasets including 15 variables as complement to already existing temperature and precipitation datasets over the recent past period. For the future period, the delta-change method was used to obtain the future climatic anomalies of precipitation and temperature. Such dataset is indeed really important as it offers us an opportunity to understand the climate dominance on other processes at higher spatial resolution and it can also facilitate the high-resolution process-based model simulation as forcing datasets. However, I find the data description mainly focused on northern hemisphere, and some sentences are just common sense, which should be revised a bit. I have a few concerns about the dataset validation and its performance in terms of the comparison with other existing available products. The validation part should be extended to include the comparison at different time scales: seasonal cycle, inter-annual variability, etc as the monthly dataset is provided.

Major comments:

1 Please describe the method you used to create the dataset in abstract as well (in one sentence) about interpolation.

We have now added a sentence detailing the downscaling procedure to the abstract.

*R1.2:*
2 The author spent a lot of words on the performance of each climate variable, but not on the process to create high spatial resolution dataset. Does the interpolation really introduce new information during the downscaling process?

Sections 2.1 and 2.2 in the methods describe whether and how the downscaling was done for preprocessing and for the BIOCLIM+ variables. Briefly, we have downscaled bias-corrected ISIMIP3b projections and *sfcWind*, using the delta-change method, and *hurs* and *clt*, using a mechanistic downscaling approach that accounts for wind and orography. The other variables were not directly downscaled but calculated based on – in addition to the aforementioned variables - previously downscaled products for temperature and precipitation (Karger et al., 2017), and solar radiation (Karger et al. submitted to ESSD), using established models and approaches. We have now indicated this more explicitly in the introductory paragraph of the methods and we have summarized the methods in more detail it in the abstract (see R1.1).

As for whether the interpolation introduces new information during downscaling, consider the example of relative humidity. Relative humidity increases when air is uplifted while being pushed over a mountain range, and it decreases when the air sinks down on the other side. In the European Alps, for example, this phenomenon is known as "foehn" wind, where typically the region south of the Alps is wet and the region north of the Alps is warm and dry, or visa-versa. Accounting for this phenomenon, our downscaling approach allows to physically constrain the assumptions about the distribution of *hurs* within an ERA5 pixel (compared to bilinear interpolation or geographically weighted regression), but of course it does not provide hard evidence about the actual fine-scale distribution of *hurs*. For most of the Earth's surface, such evidence unfortunately is not available.

*R1.3:*

Swiss Federal Institute for Forest, Snow and Landscape Research WSL
Zürcherstrasse 111, CH-8903 Birmensdorf, phone +41-44-739 21 11, fax +41-44-739 22 15, www.wsl.ch

3 Please use a table to summarize the dataset you used including the external data, specifying the spatial and temporal resolution, time period, and the source.

Table 1 now summarizes all input data used to generate the CHELSA-BIOCLIM+ data set.

*R1.4:*
4 Can you provide other evidence to justify the process of wind speed data?

We have now clarified in the subsection "Near-surface wind speed (*sfcWind*)" that we used the delta-change method to downscale wind speed. More precisely, we ran delta change on the log-transformed layers to account for the nature of the frequency distribution of wind speed (Weibull distribution).

*R1.5:*
5 The validation part should be extended, including comparison with other climate datasets not only the station-based measurement.

We have now substantially expanded the validation part. Namely we have: (1) added Figures 17 and 18, showing how validation error of our variables is distributed in space and time, respectively, and we have described these results in Lines 803-831 (see also R2.2); (2) created Figure 19, where we compared *hurs*, *sfcWind*, *vpd*, *rsds*, *pet*, and *cmi* against station-based interpolations and weather research and forecasting model outputs for the Himalaya region. These results are described in subsection 3.2.2; and (3) expanded the validation to also include alternative products building on station-based interpolations. The corresponding statistics were added to what is now Table 2.

*R1.6:*
6 I see that some figures show the difference between climatological means of 2071-2100 and 1981-2010, and others show the range of monthly means for the period 1981-2010. The former one is for variables that use temperature and precipitation only, right? You can also show range of monthly means for the common time period.

Yes, for projected variables we compare the periods 1981-2010 to 2071-2100 while for time-series variables we show seasonal variation for northern-hemisphere biomes; and yes, projected variables build on temperature and precipitation. However, we cannot show monthly means of projected variables for 1981-2010 because those are all annual statistics without monthly resolution: frost change frequency, snow cover days, potential net primary productivity, growing season length, growing season precipitation, growing degree days, and growing season temperature either have no meaning at a monthly basis or were not calculated at this temporal resolution.

It may be interesting to calculate monthly maps for *fcf*, *scd*, *npp*, and *gdd*, and we may do so in the future, but this is beyond the scope of this study. Given the >4000 global raster layers, the 15 variables, the time series, the summary statistics, and the future projections, we believe the current volume of the CHELSA-BIOCLIM+ product already meets the standard for a data set presented in a data paper.

*R1.7:*
Minor comments:

Line 33 live -> lives

Swiss Federal Institute for Forest, Snow and Landscape Research WSL
Zürcherstrasse 111, CH-8903 Birmensdorf, phone +41-44-739 21 11, fax +41-44-739 22 15, www.wsl.ch

Following R2.6, we have replaced live with life here.

*R1.8:*
Line 35 key to what

Understanding how climate change affects climate-driven processes is key to mitigate negative climate-change impacts. We have now rephrased the sentence to make this clearer.

*R1.9:*
Line 51 please add references for the usage in macroecology

We have added a reference (Fourcade et al. 2018) that demonstrates the popularity of the 19 bioclimatic variables based on a literature review.

*R1.10:*
Line 59 please use unit mm yr-1

We prefer not to, because for clarity it appears natural to consistently use the same unit. Millimeters per year are temperature dependent (as the density of water is temperature dependent) which would mean that we need a base temperature for each pixel. This would make comparisons between different climates complicated, and also comparisons between precipitation and potential evapotranspiration. Moreover, in our global grids precipitation can be solid (snow), making a reporting in mm yr$^{-1}$ even less practical. For precipitation, kg m$^{-2}$ yr$^{-1}$ is the common unit used in climate science (see e.g. ERA5, IPCC). We therefore prefer to stick with this unit.

*R1.11:*
Line 99 why the period 1981-2010 was chosen rather than 1980-2018?

Because 1981-2010 is the official 'climate normal', i.e., reference period recommended by the World Meteorological Organization (Arguez and Vose, 2011), so using it makes our data compatible with other products. We have now added this justification to the manuscript.

*R1.12:*
Line 150 to amount of

We have adapted the sentence.

*R1.13:*
Line 189 Please also mention the ERA5 spatial resolution here.

Following R1.3 we have added this information to Table 1 and fully describe the properties of all input data used in section "2.1 Input data". For consistency, we prefer not to detail input data properties again in section 2.2.

*R1.14:*
Line 231 Why is clt downscaled to 1.5 arcmin here?

*clt* has been downscaled to 1.5 arcmin in order to avoid ogrographic effects to be overrepresented. We have now justified this.

Swiss Federal Institute for Forest, Snow and Landscape Research WSL
Zürcherstrasse 111, CH-8903 Birmensdorf, phone +41-44-739 21 11, fax +41-44-739 22 15, www.wsl.ch

*R1.15:*
Line 481 is it really an east-west belt from Ethiopia to Sierra Leone?

We have replaced the statement with "along an east-west belt in subtropical Africa, roughly from the southern tip of the Red Sea to the Atlantic Ocean".

*R1.16:*
Figure 2 why did you only focus on the northern hemisphere?

Because otherwise the seasonal signals would be confounded due to the alternating seasons between hemispheres and because not all biomes exist on the southern hemisphere. Note that this is a data paper and our main aim was to demonstrate that the content of our products is sensible, rather than exhaustively describing the climate on Earth in relation to global biomes.

*R1.17:*
Figure 4 Have you checked whether the wind data also shows stilling reversal around 2010 like Zeng et al (2018)?

Thank you for hinting at this paper. As the paper mentions, "global terrestrial stilling is either not reproduced or it has been largely underestimated in global reanalysis products". This includes ERA5 (see Supplementary Figure 1 in Zheng et al. 2018) which was the basis of our *sfcWind* data. Correspondingly, patterns of wind speed stilling reversal can hardly be seen Figure 4d, and this may be a reason for the somewhat lower Pearson correlation coefficients between the monthly *sfcWind* layers and station data in recent years (Fig 18b). Despite its advantages, such as physical consistency among many climate variables, reanalysis of observational data with high-resolution weather forecast models appears to have limitations for some variables. We have now included this observation in the discussion.

Note that Zeng et al. 2018 made us aware of excellent global data of meteorological station measurements with which we have now updated our validation.

*R1.18:*
Line 560 is this potential NPP based on Miami model really meaningful? What are the benefits of this NPP dataset relative to CMIP5/6 outputs?

As we indicate in the discussion, of course the Miami model is somewhat simplistic. However, it reflects the fundamental limitations of productivity and therefore makes a powerful hypothesis of how changing climate is going to affect ecosystem productivity. In general, the expected changes according to Fig. 7b represent a fundamental expectation of how changing temperature and precipitation (CMIP5/6) outputs affect a central ecosystem function, given that we follow SSP370 and that MPI-ESM 1-2-HR generates realistic patterns. We have now added this view to the discussion.

*R1.19:*
Line 716 please also specify the comparison time period.

We have now added comparisons for various time periods and indicated them in detail.

*R1.20:*

Can you use other related (not exactly the same) variables to validate the performance of fourth-order and fifth-order climatic variables?

We have now calculated fourth-order *cmi* from FAOCLIM station data, by station-wise calculating monthly differences between *pet* and precipitation, similar to Zomer et al. (2022), and used it to validate the BIOCLIM+ variable and a station-based alternative.

*R1.21:*
Line 743-744, can you provide references more recently?

We have added more recent references.

**Detailed response to comments of Referee 2**

*R2.1:*
It is indeed very useful to develop a number of climatic variables at the interface between ecology and climate science. Such efforts can act as a link, connecting the two disciplines and helping ecologists include climatic variables into their modelling efforts, at a very high spatial resolution. Moreover, projects like this can encourage the collaboration between climate scientists and ecologists, further developing the field of "climate change ecology". Even though this study is ambitious and could potentially assist multiple ecological studies, it is lacking in terms of validation.

Major comments:

1. The fact that the data from the five climate models, and from the three scenarios (SSPs) are included independently is positive. It allows users to quantify the uncertainty introduced by these two sources (model spread, scenario uncertainty). More models (perhaps in future releases) would be even better.

We agree that more SSPs would be preferable for more nuanced analyses on the potential impact human behavior on future climate change and its impact. The associated computational and data demands, however, go beyond the scope of the data set presented here (which already consists of >4000 global raster layers). We are currently working on a software-based solution that – once ready – should allow users to downscale any ISIMIP model output (i.e., any existing bias-corrected model run for a given SSP, climate model, and period) for any study area of interest.

*R2.2:*
2. The validation presented in this paper is not convincing enough. Fifteen pages of the results section are used for describing and visualizing the variables, while only one page is devoted to validation of the data. A simple description/visualization does not add much value, and instead a comparison to the station data would have been more useful here. For example, Karger et al. (2017), compared their datasets with station data and other products using maps (spatial) and time series (temporal). This helps in gaining confidence to the CHELSA data by seing the biases over space (maps) and time (time series). For example, can two ecological studies, one in Africa and one in the Arctic, use CHELSA data with the same level of confidence? Can one study use CHELSA data for spring and for autumn with the same level of confidence?

Thanks for making this clear. We have now substantially expanded the model validation, including comparisons to other products (Table 2, Figure 19), visualizations of the spatial (Figure 17) and temporal (Figure 18) distributions of the errors, using more comprehensive validation data. Moreover, we validate an additional variable (*cmi*) and we calculate validation statistics at different temporal aggregations (climatological mean, climatological months, and monthly means).

*R2.3:*
3. It is understandable that only a number of variables (8) that are more easily measurable were validated (mostly 1st and 2nd order). It is these eight variables, after all, that are combined in order to create the other seven variables (3rd and 4th order). However, validating more than eight variables would give more confidence to the user. For example, in the case of npp, could remote sensing products be used?
https://neo.gsfc.nasa.gov/view.php?datasetId=MOD17A2_M_PSN

Our *potential* NPP builds on the Miami model and indicates how much productivity would be possible if the soil is OK and well-adapted species in a productive life stage are present. It is expected to deviate substantially from remotely-sensed NPP, especially in urban, agricultural, and probably even semi-natural regions but also in natural ecosystems after disturbance, where the soil is limiting (e.g., wetlands), or when annual climate deviates from the long-term mean. We therefore do not believe that the association between potential and observed NPP is an informative validation criterion. However, we have now derived station-based estimates for the validation of *cmi* (see R1.20).

*R2.4:*
4. Only a global value for r, RMSE, MAE and bias is provided. What about different regions or different seasons? Do all regions/seasons show similar spatial and temporal patterns?

We now provide values for *r*, RMSE, MAE, and bias for climatological means, climatological months, and – where possible - monthly means (Table 2). Moreover, we show the spatial distribution of MAE (Fig. 17) and the temporal distribution of *r* (Fig. 18).

*R2.5:*
5. Throughout the paper, the words "high spatiotemporal resolution" are used. The spatial resolution is definitely high. Is the temporal resolution of this dataset (monthly) considered high for ecological applications?

For macroecological applications we would say so. Of course "high" is always relative, but as far as we are aware, beyond CHELSA no global, climate-related products with comparable spatial resolution exist that are resolved as time series. In ecology, monthly climatologies representing 30-year averages are typically used at the global scale. Therefore, even though monthly resolution may sound low to investigate some extreme events (See R2.7), to study macroecological patterns and processes, such as seasonal phenologies or annual variations in growth, it is suitable and provides unprecedented detail. Note that most ecological process models can be operated easily with monthly time series input.

As an additional note, we would like to highlight that establishing maintaining the IT infrastructure and storage capacity required to offer global layers at e.g. daily resolution is challenging and costly, especially for the large number of variables included in this data set. We have done so for precipitation and *rsds* elsewhere (Karger et al., 2021, Karger et al. in prep). In this work, however, our focus was establishing an extensive set of relevant predictors and offer them at unprecedented resolution. To keep the data volume manageable, we could not go

below monthly resolution. To provide customizable data at higher resolution, we are currently working on a software solution (see also R2.1).

*R2.6:*
Minor comments:

Line 33: live -> life

Done.

*R2.7:*
Line 35: This dataset's highest temporal resolution is monthly, is this enough for detecting extremes? Please provide examples and/or references

Thanks for the hint. We agree that a monthly temporal resolution may be too crude to investigate floods and therefore have now limited the focus on droughts in the example.

*R2.8:*
Line 71: Same comment as in Line 35

We believe that for large-scale analyses monthly resolution is sufficient to identify drought and link it to disturbances. This may, however, be different for regional studies for which station-based estimates of temperature and precipitation anomalies may be better suited (see e.g. Brun et al., 2020). We have now specified in the corresponding sentence that we refer to large-scale analyses.

*R2.9:*
Line 80: Please provide references

We have now added three references to support the statements made in the sentence.

*R2.10:*
Line 447: Please explain how r, MAE, RMSE and bias are calculated. Did you use time series from each station, and then averaged them globally to obtain a single number (the number shown in Table 1)?

We have rewritten the methods section on validation and now explicitly describe how the temporal aggregation of station data was done. Moreover, we now validate the data at different temporal aggregations.

*R2.11:*
Line 732: Please provide references for "Moreover, they are comparably robust to gaps in the network of meteorological field stations and therefore provide more reliable estimates in remote areas, compared to alternatives from station-based interpolations"

Thanks for the hint. As it was, the statement made too much interpretation without references. We have now removed it from the first paragraph of the discussion (which now only summarizes the results) and elaborated on robustness in remote areas, with more references, in the dedicated paragraph below (Lines 866-878). Moreover, Figure 19 in the results now also provides an impression of the patterns of BIOCLIM+ variables relative to other products in the remote Himalaya region.

*R2.12:*
Line 762: "... realistic patterns in our exemplary remote-area, high-resolution maps in ...".
Realistic compared to what? Please define "realistic" here.

We have now removed the sentence and justify our expectation that the product performs well in remote areas with experiences of ecological studies that compared station-based interpolations of temperature and precipitation with CHELSA. CHELSA temperature and precipitation use related downscaling approaches and partially or fully underlie most of the variables presented here.

**Bibliography**

Arguez, A. and Vose, R. S.: The Definition of the Standard WMO Climate Normal: The Key to Deriving Alternative Climate Normals, Bull. Am. Meteorol. Soc., 92, 699–704, https://doi.org/10.1175/2010BAMS2955.1, 2011.

Brun, P., Psomas, A., Ginzler, C., Thuiller, W., Zappa, M., and Zimmermann, N. E.: Large-scale early-wilting response of Central European forests to the 2018 extreme drought, Glob. Chang. Biol., 1–15, https://doi.org/10.1111/gcb.15360, 2020.

Karger, D. N., Conrad, O., Böhner, J., Kawohl, T., Kreft, H., Soria-Auza, R. W., Zimmermann, N. E., Linder, H. P., and Kessler, M.: Data from: Climatologies at high resolution for the earth's land surface areas, https://doi.org/10.5061/dryad.kd1d4, 2017.

Karger, D. N., Lange, S., Hari, C., Reyer, C. P. O., Conrad, O., Zimmermann, N. E., and Frieler, K.: CHELSA-W5E5: Daily 1 km meteorological forcing data for climate impact studies, submitted to Earth System Science Data on Nov 3, 2022.

Karger, D. N., Wilson, A. M., Mahony, C., Zimmermann, N. E., and Jetz, W.: Global daily 1 km land surface precipitation based on cloud cover-informed downscaling, Sci. Data, 8, 307, https://doi.org/10.1038/s41597-021-01084-6, 2021.

Zomer, R. J., Xu, J., and Trabucco, A.: Version 3 of the Global Aridity Index and Potential Evapotranspiration Database, Sci. Data, 9, 409, https://doi.org/10.1038/s41597-022-01493-1, 2022.